# Isoform-specific disruption of the *TP73* gene reveals a critical role for TAp73γ in tumorigenesis via leptin

Xiangmudong Kong[1], Wensheng Yan[1†], Wenqiang Sun[1], Yanhong Zhang[1], Hee Jung Yang[1‡], Mingyi Chen[2], Hongwu Chen[3], Ralph W de Vere White[4], Jin Zhang[1]*, Xinbin Chen[1]*

[1]Comparative Oncology Laboratory, Schools of Veterinary Medicine and Medicine, University of California, Davis, Davis, United States; [2]Department of Pathology, University of Texas Southwestern Medical Center, Dallas, United States; [3]Department of Biochemistry and Molecular Medicine, University of California, Davis, Davis, United States; [4]Department of Urology Surgery, School of Medicine, University of California, Davis, Davis, United States

**\*For correspondence:**
jinzhang@ucdavis.edu (JZ);
xbchen@ucdavis.edu (XC)

**Present address:** [†]Berkeley Regional Lab, Pathology/ Lab-Histology Department, The Permanente Medical group, Berkeley, United States; [‡]Pharmacology Team, Drug Discovery Center, LG Life Science R&D, LG Chem Ltd, Seoul, South Korea

**Abstract** *TP73*, a member of the p53 family, is expressed as TAp73 and ΔNp73 along with multiple C-terminal isoforms (α−η). ΔNp73 is primarily expressed in neuronal cells and necessary for neuronal development. Interestingly, while TAp73α is a tumor suppressor and predominantly expressed in normal cells, TAp73 is found to be frequently altered in human cancers, suggesting a role of TAp73 C-terminal isoforms in tumorigenesis. To test this, the TCGA SpliceSeq database was searched and showed that exon 11 (E11) exclusion occurs frequently in several human cancers. We also found that p73α to p73γ isoform switch resulting from E11 skipping occurs frequently in human prostate cancers and dog lymphomas. To determine whether p73α to p73γ isoform switch plays a role in tumorigenesis, CRISPR technology was used to generate multiple cancer cell lines and a mouse model in that *Trp73* E11 is deleted. Surprisingly, we found that in E11-deificient cells, p73γ becomes the predominant isoform and exerts oncogenic activities by promoting cell proliferation and migration. In line with this, E11-deficient mice were more prone to obesity and B-cell lymphomas, indicating a unique role of p73γ in lipid metabolism and tumorigenesis. Additionally, we found that E11-deficient mice phenocopies *Trp73*-deficient mice with short lifespan, infertility, and chronic inflammation. Mechanistically, we showed that Leptin, a pleiotropic adipocytokine involved in energy metabolism and oncogenesis, was highly induced by p73γ, necessary for p73γ-mediated oncogenic activity, and associated with p73α to γ isoform switch in human prostate cancer and dog lymphoma. Finally, we showed that E11-knockout promoted, whereas knockdown of p73γ or Leptin suppressed, xenograft growth in mice. Our study indicates that the p73γ-Leptin pathway promotes tumorigenesis and alters lipid metabolism, which may be targeted for cancer management.

## Editor's evaluation

TP73 is a member of the p53 family of tumor suppressors and is expressed as TAp73 and DNp73 and multiple C-terminal isoforms as a result of alternative splicing. This manuscript describes an interesting study revealing the complex and intricate functional network driven by p73 isoforms. Using elegant in vitro and in vivo assays the authors provide compelling evidence that a TAp73-α to TAp73-γ switch could be a frequent phenomenon in human cancers and provide novel evidence that TAp73-γ has oncogenic functions via Leptin.

## Introduction

The *TP73* gene encodes a member of the p53 family of transcription factors involved in tumor suppression and development. *TP73* gene is mapped to a region on human chromosome 1p36 that is frequently deleted in neuroblastoma and other tumors, thus linking its role to cancer (*Kaghad et al., 1997*). p73 shares a high similarity with p53 in the transactivation, tetramerization, and DNA-binding domains, with most considerable homology between their DNA-binding domains (*Harms et al., 2004*). Consequently, p73 activates a number of putative p53 target genes involved in cell cycle arrest, apoptosis, and differentiation (*Harms et al., 2004*; *Ichimiya et al., 2000*). Despite these similarities, p73 is not functionally redundant to p53. *TP73* is rarely mutated but frequently found to be upregulated in human cancers, indicating that p73 is not a classic Knudson-type tumor suppressor (*Ozaki and Nakagawara, 2005*; *Nomoto et al., 1998*; *Nimura et al., 1998*). Indeed, the biological functions of p73 have been linked to neurological development, tumorigenesis, ciliogenesis, fertility, and metabolism as illustrated by in vitro and in vivo models (*Yang et al., 2000*; *Nemajerova et al., 2016*; *Marshall et al., 2016*; *Tissir et al., 2009*; *Wilhelm et al., 2010*; *Tomasini et al., 2008*; *Rufini et al., 2012*; *Inoue et al., 2014*; *Dulloo et al., 2015b*).

The role of p73 in cancer is complicated due to the presence of multiple p73 isoforms that coordinate in a complex way to exert both oncogenic and non-oncogenic activities (*Dulloo et al., 2015b*; *Nemajerova et al., 2018*; *Rufini et al., 2011*; *Stiewe and Pützer, 2002*; *Amelio et al., 2015*). Through the usage of two promoters, *TP73* encodes two classes of isoforms, TAp73 and N-terminally truncated ΔNp73. TAp73 isoforms, driven by the P1 promoter located upstream of exon 1, contain a transactivation domain similar to that in p53 and thus are suggested to function as a tumor suppressor. ΔNp73 isoforms, arising from the P2 promoter located in intron 3, lack the N-terminal transactivation domain, and thus, are thought to have oncogenic potential by acting as dominant-negative inhibitors of p53 and TAp73. Mice deficient in ΔNp73 isoforms are not prone to tumors, but have defects in neurological development similar to that in total Trp73-KO mice (*Tissir et al., 2009*; *Wilhelm et al., 2010*). By contrast, mice deficient in TAp73 isoforms develop spontaneous and carcinogens-induced tumors (*Tomasini et al., 2008*), suggesting a role of TAp73 as tumor suppressor. However, several recent studies also showed that TAp73 exhibits both pro-tumorigenic activities depend on the context (*Dulloo et al., 2015b*; *Li et al., 2018*; *Sharif et al., 2019*), suggesting that the role of TAp73 in tumorigenesis is far more complex, likely being affected by the various C-terminal TAp73 isoforms.

At the C-terminus, multiple p73 isoforms are generated due to alternative splicing from exons 11–13 (*Murray-Zmijewski et al., 2006*; *Vikhreva et al., 2018*). At transcript levels, six splicing variants (α, β, γ, δ, ε, ζ) can be detected. However, at protein levels, only three isoforms (α, β, and γ) can be detected, with α being the most abundantly expressed. Due to lack of proper model systems and reagents, most studies about p73 C-terminal isoforms have been done in cell culture and, thus, may not reflect the authentic biological role of these isoforms. When overexpressed, p73β showed very strong, whereas p73α and p73γ exhibited much weaker, activity to induce programmed cell death (*Liu and Chen, 2005*; *Ueda et al., 1999*). These differential activities are suggested due to the variation of C-terminal sequence among p73 C-terminal isoforms (*Vikhreva et al., 2018*). p73α, the longest isoform, has 636 aa and contains a sterile alpha motif (SAM) and an extreme C-terminal domain, both of which are thought to inhibit its transcriptional activity through interaction with other factors (*Ozaki et al., 1999*). p73γ is produced from splicing out of exon 11, which creates a long alternative reading frame that leads early stop codon. As a result, TAp73γ has 475 aa and contains a unique C-terminal domain (the last 76 aa) with unknown function. By contrast, p73β does not contain an extended C-terminal domain present in both p73α and p73γ. Nevertheless, these hypotheses have never been tested in vivo. The difficulties to study the physiological roles of p73 C-terminal isoforms are mainly due to the following reasons: (1) the presence of TA/ΔN isoforms with opposing functions, which adds complications to study the C-terminal isoforms; (2) p73α is the major isoform expressed in most cells and tissues and may shield activities by other isoforms; and (3) lack of specific antibodies to detect β or γ isoforms. These limitations may have led to two opposing functions for TAp73 as a tumor suppressor and a tumor promoter (*Sabapathy, 2015*).

Leptin is a 167-amino-acid peptide hormone encoded by the *Ob* gene and produced by a multitude of tissues, especially adipose tissue (*Zhang et al., 1994*). Leptin communicates with its receptor (LepR) in the hypothalamus and controls energy expenditure and appetite through a negative feedback loop, thereby maintaining the relative constancy of adipose tissue mass from being too thin or

too obese (*Coleman, 2010*; *Friedman, 2019*). The anti-obesity effect of Leptin was demonstrated in individuals bearing congenital leptin deficiencies (*Farooqi et al., 1999*). It was later found out that leptin therapy is ineffective as most obese individuals have a high level of endogenous plasma leptin (hyperleptinemia) and do not respond to leptin therapy, a condition called leptin resistance (*Frederich et al., 1995*). As a pleiotropic cytokine, leptin plays a critical role in lipid/glucose homeostasis, immune responses, hematopoiesis, angiogenesis, reproduction, and mental processes (*Bennett et al., 1996*; *Chehab et al., 1996*; *Ducy et al., 2000*; *Sierra-Honigmann et al., 1998*). Leptin signaling is also found to promote cancer progression, including cell proliferation, metastasis, angiogenesis, and chemoresistance (*Ray and Cleary, 2017*; *Xu et al., 2020*). Notably, both leptin and LepR are frequently altered in many obesity-associated cancers, including lung, breast, ovarian, pancreas, liver, and prostate (*Hosney et al., 2017*; *Lin and Hsiao, 2021*; *Karabulut et al., 2016*). Thus, targeting Leptin-LepR signaling may represent a novel strategy for cancer therapeutics.

To understand the role of various p73 C-terminal isoforms in cancer development, we searched the TCGA SpliceSeq database and found that *Trp73* E11 skipping, which led to isoform switch from p73α to p73γ, frequently occurs in a subset of human cancers and dog lymphomas. Thus, to explore the biological function of p73γ isoform, we employed CRISPR technology to manipulate p73 splicing by deleting E11 in the *TP73* gene in multiple cancer cell lines and mice. We showed that E11 deficiency resulted in p73γ becoming the predominant isoform in both human and mouse cells. Unexpectedly, we found that p73γ, when expressed at a physiologically relevant level, did not behave as a tumor suppressor, but promoted cell migration and tumorigenesis in vitro and in vivo. Mechanistically, we showed that Leptin, a pleiotropic adipocytokine, was induced by p73γ and contributes to p73γ-mediated tumorigenesis. Moreover, we found that p73α to γ isoform switch was detected in a subset of dog lymphomas and human prostate carcinomas along with elevated expression of Leptin. Finally, we showed that targeting p73γ or Leptin led to growth suppression of E11-KO xenografts in a mouse xenograft model, suggesting that the p73γ-Leptin pathway may be explored for cancer management.

## Results

### p73α-γ switch is detected in a subset of human prostate carcinomas and dog lymphomas

Early studies have shown that p73 C-terminal isoforms are dysregulated in breast cancer and leukemia (*Rufini et al., 2011*; *Zaika et al., 1999*). However, it is not clear how p73 C-terminal isoforms are altered and whether these alterations play a role in tumorigenesis. In this regard, we examined the splicing pattern of *TP73* in normal and tumor tissues by using the TCGA SpliceSeq database. Interestingly, we found that the percentage of exon 11 exclusion was much higher in lung squamous cell carcinomas (LUSCs) and head-neck squamous cell carcinomas (HNSCs) when compared to their respective normal tissues (*Figure 1A and B*). We would like to note that exon 11 skipping would lead to production of two p73 isoforms: p73γ, which is switched from p73α, and p73ε, which is switched from p73β. Thus, to determine which p73 isoform is altered in cancer tissues, expression of *p73α*, *p73γ*, and *p73ε* transcripts was measured in 5 normal human prostates and 16 prostate carcinomas. We found that *p73α* was detectable in 5 normal prostates, undetectable in 12 prostate carcinomas, and unaltered or slightly elevated in 4 prostate carcinomas (*Figure 1C and D*). By contrast, *p73γ* was highly expressed in prostate carcinomas compared to that in normal prostates (*Figure 1C and D*, compare lanes 1–5 with 6–13). Additionally, *p73ε* was almost undetectable at both normal and prostate cancer tissues (data not shown). Similarly, we found that *p73α* was expressed in 3 normal dog lymph nodes but undetectable in 16 dog lymphomas (*Figure 1E*, *p73α* panel, compare lanes 1–3 with 4–16). By contrast, p73γ expression was very low in 3 normal dog lymph nodes but elevated in all 16 dog lymphomas (*Figure 1E*, *p73γ* panel, compare lanes 1–3 with 4–16). These data suggest that *p73α* isoform in normal tissues is switched to p73γ isoform in cancer tissues and that *p73γ* is associated with tumorigenesis.

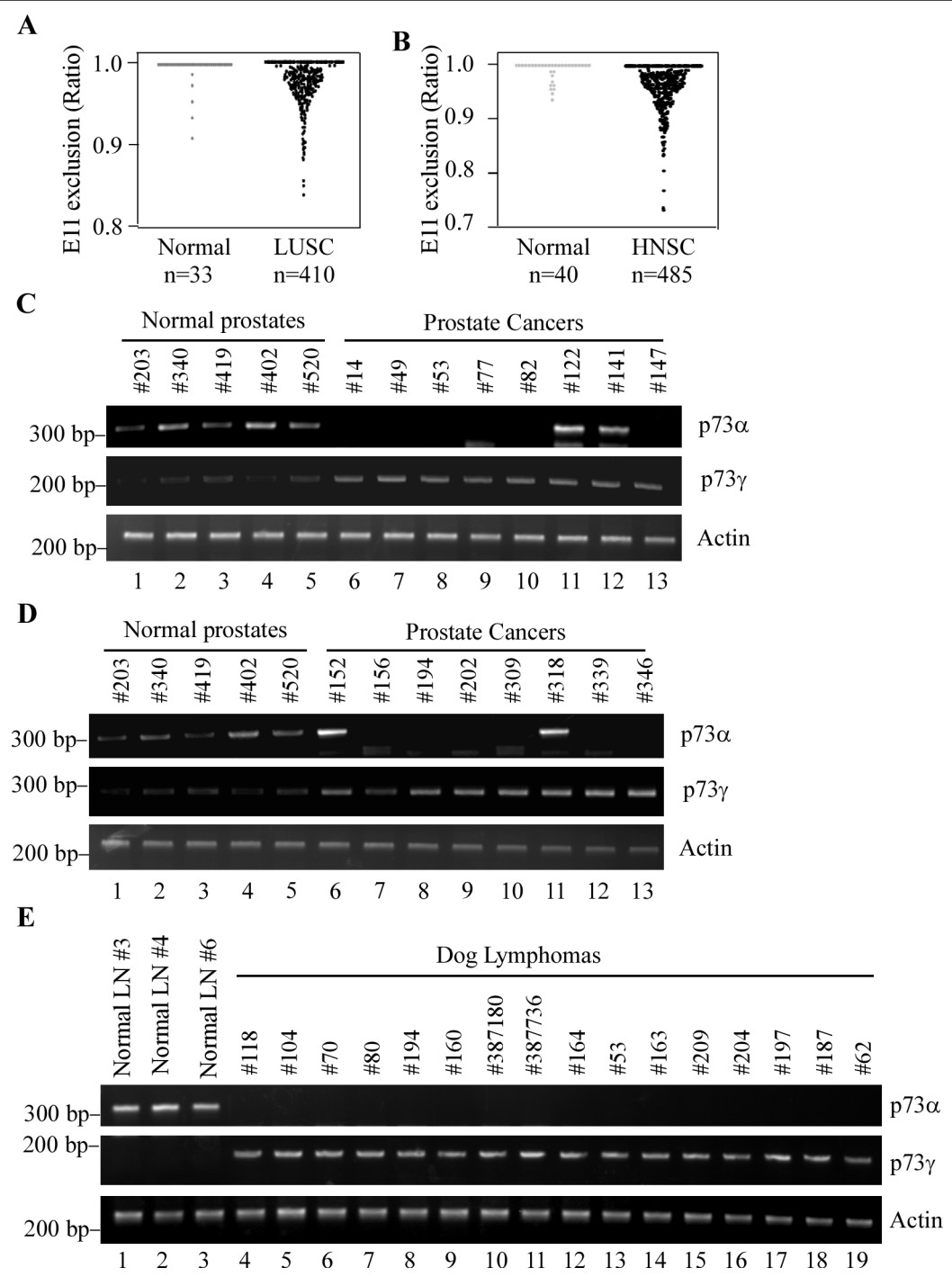

**Figure 1.** E11 skipping is detected in a subset of human cancers and dog lymphomas. (**A, B**) Ratio of *TP73* exon 11 exclusion was examined in lung squamous cell carcinoma (LUSC) and head-neck squamous cell carcinoma (HNSC) along with their normal tissues by using TCGA SpliceSeq database. Value of 1 indicates 100% of the transcripts contains all exons, for example, p73α. Values less than 1 represents the extent of expression of the isoforms with exon 11 exclusion, for example, *p73γ*. (**C, D**) The levels of *p73α*, *p73γ*, and *actin* transcript were examined in 5 normal prostates and 16 prostate cancer samples. (**E**) The levels of *p73α*, *p73γ*, and *actin* transcript were examined in 3 normal dog lymph nodes and 16 dog lymphomas.

The online version of this article includes the following source data for figure 1:

**Source data 1.** Labeled gel images.

**Source data 2.** Raw gel images.

## Deletion of exon 11 in the *TP73* gene leads to isoform switch from α to γ, resulting in enhanced cell proliferation and migration as well as altered epithelial morphogenesis

To mimic E11 skipping in human cancer tissues and understand the biological functions of p73γ, CRISPR-Cas9 technology was used to generate stable H1299 and Mia-PaCa2 cell lines in that the splicing acceptor for *TP73* exon 11 (E11) was deleted by using two guide RNAs (*Figure 2—figure supplement 1A*). Sequence analysis verified that E11-KO H1299 and Mia-PaCa2 cells contained a deletion of 21 nt in intron 10 and 26 nt in E11 in the *TP73* gene. We predicted that deletion of E11 splicing acceptor would trigger E11 skipping and subsequently result in C-terminal isoform switching (*Figure 2A*). Indeed, we found that when compared to isogenic control cells, E11-KO cells expressed no α/β isoforms, increased levels of γ/ε isoforms, and unaltered δ/ζ isoforms (*Figure 2B*). Additionally, TA/ΔNp73 isoforms were expressed at similar levels in both isogenic control and E11-KO H1299 cells (*Figure 2B*, TA/ΔNp73 panels). The isoform switch in E11-KO cells was also confirmed with specific primers that amplify α/γ or β/ε isoforms (*Figure 2—figure supplement 1B and C*).

Due to lack of specific antibodies and relative low abundance, p73γ is almost undetectable under normal conditions. Thus, a peptide (PRDAQQPWPRSASQRRDE) derived from the unique C-terminal region in p73γ/ε was used to generate an antibody, called anti-p73γ/ε antibody, which was found to recognize p73γ/ε but not p73α (*Figure 2—figure supplement 1D*). We showed that in isogenic control cells TAp73α was the mostly abundant protein expressed as detected by an anti-TAp73 antibody, whereas in E11-KO cells, TAp73γ became the predominantly expressed isoform and was detected by anti-p73γ/ε antibody under normal and DNA-damage induced conditions (*Figure 2C* and *Figure 2—figure supplement 1E*). Nevertheless, other p73 isoforms, including p73β and p73ε, remained undetectable likely due to their low abundance. These data confirmed that deletion of the acceptor site for E11 in *TP73* led to E11 skipping and subsequently resulted in isoform switch from p73α in isogenic control cells to p73γ in E11-KO cells.

By using two different promoters, p73 C-terminal isoforms are expressed as TA/ΔNp73 proteins, which are known to have opposing functions (*Rufini et al., 2011*). Thus, when studying the p73 C-terminal isoforms, it is also important to consider the N-terminal variations. To this end, stable TAp73- or ΔNp73-KO H1299 and Mia-PaCa2 cell were generated (*Figure 2D* and *Figure 2—figure supplement 1F*) and used as controls to study the function of E11 deficiency. We found that knockout of TAp73 enhanced, whereas knockout of ΔNp73 reduced, colony formation compared to isogenic control cells (*Figure 2E*), consistent with previous reports (*Zhang et al., 2013*; *Zhang et al., 2012*). Interestingly, the number of colonies formed by E11-KO cells was higher than that by TAp73-KO cells, becoming the highest among all four cell lines (*Figure 2E*). To verify this, tumor sphere formation assay was performed and showed that loss of TAp73 or E11 promoted, whereas loss of ΔNp73 inhibited, cell proliferation in H1299 cells (*Figure 2—figure supplement 1G*). Moreover, to examine whether E11 deficiency affects cell migration, scratch assay was performed. We found that knockout of TAp73 promoted, whereas knockout of ΔNp73 slightly inhibited, cell migration in H1299 and Mia-PaCa2 cells in a time-dependent manner (*Figure 2F* and *Figure 2—figure supplement 1H*), consistent with previous observations (*Zhang et al., 2012*; *Bae et al., 2018*). Notably, E11-KO cells migrated even faster than isogenic control or TAp73-KO H1299 and Mia-PaCa2 cells (*Figure 2F* and *Figure 2—figure supplement 1H*). Consistently, transwell migration assay showed that E11-KO H1299 cells exhibited a markedly enhanced ability to transmigrate through a membrane compared to isogenic control, TAp73-KO, and ΔNp73-KO H1299 cells (*Figure 2—figure supplement 1I*). Furthermore, to determine whether E11 deficiency has an effect on epithelial morphogenesis, E11-KO, TAp73-KO, and ΔNp73-KO MCF10A cells were generated (*Figure 2—figure supplement 1J*) and then subjected to three-dimensional (3-D) culture. We found that parental and isogenic control MCF10A cells underwent normal cell morphogenesis (acini with hollow lumen) (*Figure 2G*, left two panels), whereas ΔNp73-KO showed delayed acinus formation (small acini) (*Figure 2G*, ΔNp73 panel). Interestingly, E11-KO and TAp73-KO showed aberrant cell morphogenesis (irregular acini with filled lumen) (*Figure 2G*, TAp73-KO and E11-KO panels). These data suggest that loss of E11 promotes oncogenesis by altering epithelial cell morphogenesis and enhancing cell proliferation and migration.

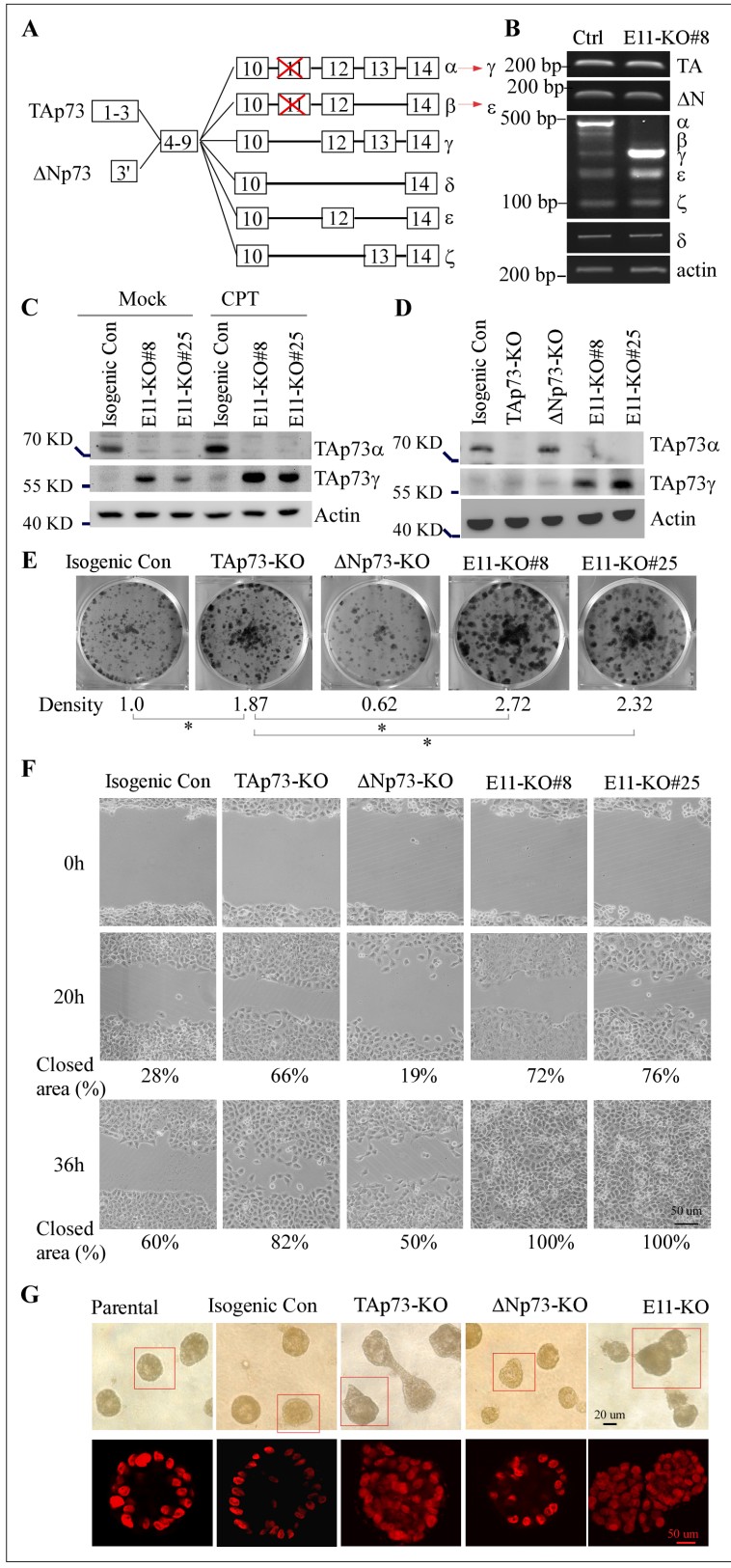

**Figure 2.** Deletion of exon 11 in the *TP73* gene leads to isoform switch from α to γ, resulting in enhanced cell proliferation and migration as well as altered epithelial morphogenesis. (**A**) Schematic representation of p73 isoforms and isoform switch resulting from E11 deletion. (**B**) The level of p73 splicing variants was examined by RT-PCR in isogenic control and E11-KO H1299 cells. (**C**) The levels of TAp73α, TAp73γ, and actin protein were

*Figure 2 continued on next page*

*Figure 2 continued*

measured in isogenic control and E11-KO H1299 cells mock-treated or treated with camptothecin (CPT). (**D**) The levels of TAp73α, TAp73γ, and actin protein were measured in isogenic control, TAp73-KO, ΔNp73-KO, and E11-KO H1299 cells. (**E**) Colony formation was performed with isogenic control, TAp73-KO, ΔNp73-KO, and E11-KO H1299 cells. The was quantified as relative density. (**F**) Scratch assay was performed with isogenic control, TAp73-KO, ΔNp73-KO, and E11-KO H1299 cells. Phase-contrast photomicrographs were taken immediately after scratch (0 hr), 20 hr, and 36 hr later to monitor cell migration. (**G**) Representative images of parental, isogenic control, TAp73-KO,ΔNp73-KO, and E11-KO MCF10A cells in three-dimensional culture. Top panel: phase-contrast image of acini in 3-D culture. Bottom panel: confocal images of cross-sections through the middle of acini stained with TO-PRO-3.

The online version of this article includes the following source data and figure supplement(s) for figure 2:

**Source data 1.** Labeled gel images.

**Source data 2.** Raw gel images.

**Figure supplement 1.** Deletion of E11 in TP73 leads to enhanced TAp73gamma expression and subseuqntly, enhanced cell proliferation and migration.

**Figure supplement 1—source data 1.** Labeled gel images.

**Figure supplement 1—source data 2.** Raw gel images.

**Figure supplement 1—source data 3.** Numerical data for *Figure 2—figure supplement 1G*.

**Figure supplement 2.** Ectopic TAp73gamma promotes cell migration.

**Figure supplement 2—source data 1.** Labeled gel images.

**Figure supplement 2—source data 2.** Raw gel images.

## TAp73γ is primarily responsible for the oncogenic effects observed in E11-KO cells

The enhanced potential of cell proliferation and migration in E11-KO cells could be mediated by increased TAp73γ expression or simply by loss of TAp73α as similar phenotypes were also observed in TAp73-KO cells albeit to a less extent (*Figure 2E–G* and *Figure 2—figure supplement 1G–I*). Thus, to characterize the role of TAp73γ in oncogenesis, stable H1299 cell lines in that TAp73γ can be inducibly expressed under the control of a tetracycline-inducible promoter were generated (*Figure 2—figure supplement 2A*). We showed that upon induction TAp73γ promoted cell migration (*Figure 2—figure supplement 2B*), consistent with the data from E11-KO cells (*Figure 2F* and *Figure 2—figure supplement 1G–I*). Next, to determine the role of endogenous p73γ in cell proliferation and migration, two p73α/γ siRNAs (si-p73α/γ#1 or #2), which were designed to target the junction region of E12-E13 or E13-E14 in *p73α/γ* mRNA (*Figure 3—figure supplement 1A*), were synthesized and found to efficiently knock down p73α and p73γ in isogenic control and E11-KO cells (*Figure 3A and B*). Next, cell migration was measured by scratch and transwell assays. We showed that knockdown of p73α/γ enhanced cell migration in isogenic control cells (*Figure 3C* and *Figure 3—figure supplement 1B*). By contrast, knockdown of p73α/γ decreased cell migration in E11-KO cells (*Figure 3C* and *Figure 3—figure supplement 1B*). Similarly, we found that knockdown of p73α/γ enhanced cell viability of tumor spheres in isogenic control cells but decreased in E11-KO cells (*Figure 3—figure supplement 1C*). These data suggested that cell proliferation and migration is inhibited by p73α, the major isoform in isogenic control cells, but enhanced by p73γ, the major isoform in E11-KO cells.

p73γ can be expressed as two isoforms, TAp73γ and ΔNp73γ. To determine which p73γ isoform is responsible for the enhanced cell migration in E11-KO cells, we generated a dual-KO (DKO) cell line by deleting all ΔNp73 isoforms in E11-KO H1299 and Mia-PaCa2 cells. We showed that ΔNp73 transcript was not expressed, whereas a similar amount of TAp73γ protein was expressed, in ΔNp73/E11-DKO H1299 and Mia-PaCa2 cells as compared to that in E11-KO cells (*Figure 3D and E* and *Figure 3—figure supplement 1D and E*). We found that the ability of cells to migrate was not altered by ΔNp73/E11-DKO as compared to that by E11-KO (*Figure 3F* and *Figure 3—figure supplement 1F*). Considering that knockout of ΔNp73 inhibited cell proliferation and migration (*Figure 2E and F* and *Figure 2—figure supplement 1G–H*), we conclude that TAp73γ was able to overcome the inhibitory effect of ΔNp73 deficiency on cell proliferation and migration. Together, these data suggest that

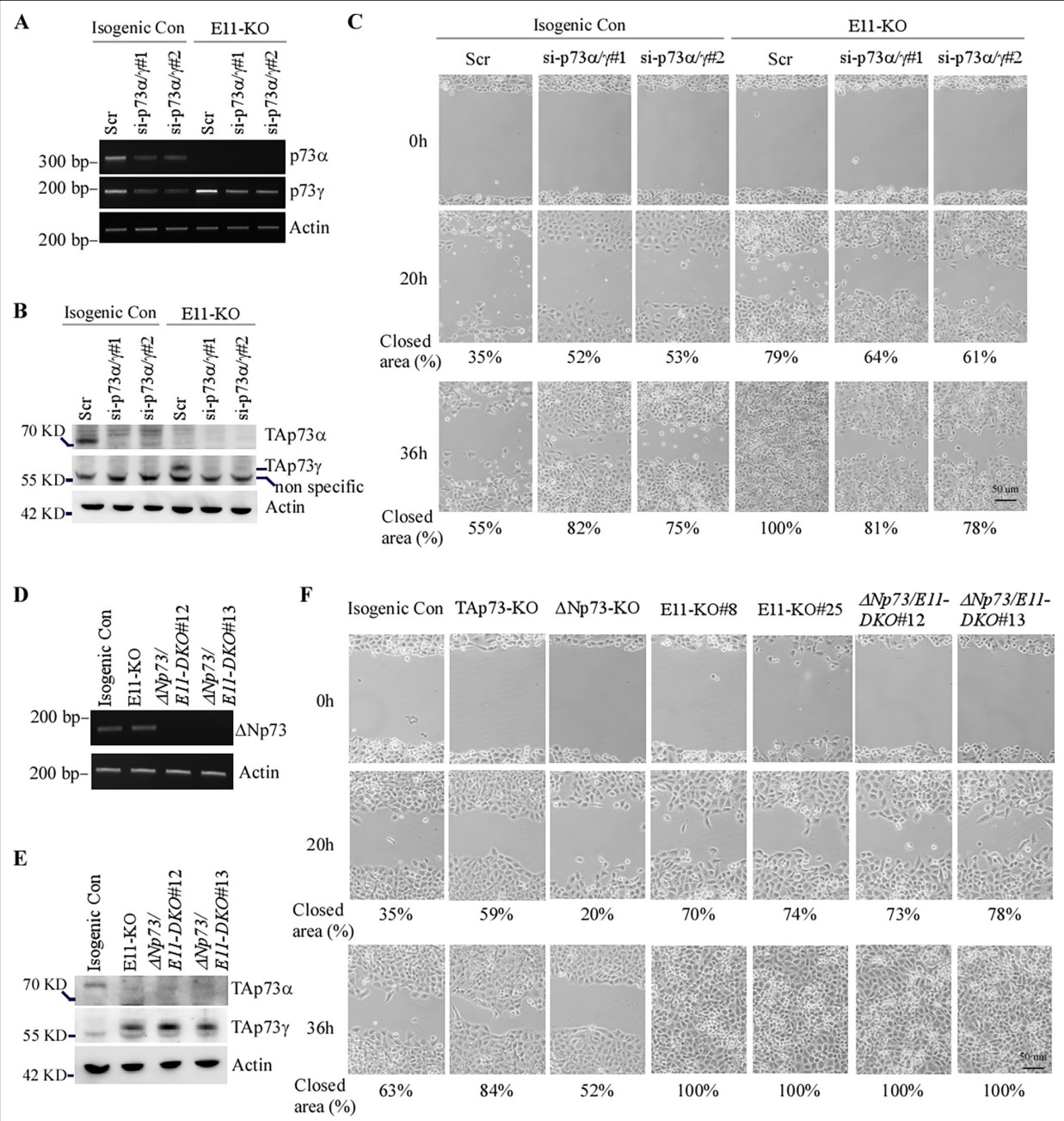

**Figure 3.** TAp73γ is primarily responsible for the oncogenic effects observed in E11-KO cells. (**A**) Isogenic control and E11-KO H1299 cells transfected with a scrambled siRNA or siRNAs against p73α/γ for 3 d, followed by RT-PCR analysis to measure the levels of *p73α*, *p73γ*, and *actin* transcripts. (**B**) Cell were treated as in (**A**) and then subjected to western blot analysis to measure the levels of TAp73α, TAp73γ, and actin proteins using antibodies against TAp73, p73α/γ, and actin. (**C**) Scratch assay was performed with isogenic control and E11-KO H1299 cells transfected with a scrambled siRNA or siRNAs against p73α/γ for 3 d. Phase-contrast photomicrographs were taken immediately after scratch (0 hr), 20 hr, and 36 hr later to monitor cell migration. (**D**) The levels of *ΔNp73* and *actin* transcripts were measured in isogenic control, E11-KO, ΔNp73/E11-DKO H1299 cells. (**E**) The levels of TAp73α, TAp73γ, and actin proteins were measured by western blot analysis using isogenic control, E11-KO, ΔNp73/E11-DKO H1299 cells. (**F**) Scratch assay was performed with isogenic control, TAp73-KO, ΔNp73-KO, E11-KO, and ΔNp73/E11-DKO H1299 cells. Phase-contrast photomicrographs were taken immediately after scratch (0 hr), 20 hr, and 36 hr later to monitor cell migration.

The online version of this article includes the following source data and figure supplement(s) for figure 3:

**Source data 1.** Labeled gel images.

**Source data 2.** Raw gel images.

**Figure supplement 1.** TAp73gamma, but not DNp73 is required for the enhanced cell mgiration and proliferation.

*Figure 3 continued on next page*

*Figure 3 continued*

**Figure supplement 1—source data 1.** Labeled gel images.

**Figure supplement 1—source data 2.** Raw gel images.

**Figure supplement 1—source data 3.** Numerical data for *Figure 3—figure supplement 1C*.

TAp73γ is primarily responsible for the enhanced cell proliferation and migration observed in E11-KO cells.

## Deletion of E11 in the *Trp73* gene leads to shortened lifespan, increased incidence of spontaneous tumors and chronic inflammation in mice

CRISPR-Cas9 was used to generate a mouse model in that E11 in the *Trp73* gene was deleted (*Figure 4—figure supplement 1A*). A cohort of WT, E11-HET, and E11-KO MEFs was then generated (*Figure 4—figure supplement 1B*) and used to examine whether E11 deficiency would lead to isoform switch in mice. We found that although MEFs mainly expressed *p73α/γ/ζ* transcripts, E11 deficiency led to isoform switch from p73α in WT MEFs to p73γ in both E11-HET and E11-KO MEFs (*Figure 4A*), which was similar to that in human cells (*Figure 2B*). Next, SA-β-gal staining was performed to evaluate whether E11 deficiency had an effect on cellular senescence, an intrinsic mechanism of tumor suppression. Surprisingly, E11-KO MEFs were less prone to cellular senescence compared to WT MEFs (*Figure 4B*), along with decreased expression of senescence markers, including PML, p130, and p21 (*Figure 4C*).

To examine the biological functions of E11 deficiency in vivo, a cohort of E11-Het and E11-KO mice was generated. We found that due to loss of TAp73α, E11-KO mice were runty, infertile, and prone to hydrocephalus (*Figure 4—figure supplement 1C* and data not shown). These observations from E11-KO mice are similar to the phenotypes from *Trp73*[-/-] mice, which also lacks of TAp73α (*Yang et al., 2000*; *Inoue et al., 2014*). Thus, E11-KO mice were not suitable for long-term tumor study. However, young E11-HET mice appeared to be normal. In this regard, a cohort of E11-HET mice was generated and monitored for potential abnormalities along with WT and *Trp73*[+/-] mice. We would like to mention that to minimize the number of animals used, all WT and 26 out of 30 *Trp73*[+/-]mice, which were generated previously but had same genetic background and maintained in the same facility (*Zhang et al., 2017*; *Zhang et al., 2014*; *Zhang et al., 2019*), were used as controls. We found that the lifespan for E11-HET mice (96 wk) and *Trp73*[+/-] mice (88 wk) was much shorter than that for WT mice (117 wk) (*Figure 4D*). However, there was no difference in lifespan between *Trp73*[+/-] and E11-HET mice (*Figure 4D*). Pathological analysis indicated that 11 out of 51 WT mice, 13 out of 27 *Trp73*[+/-] mice, and 17 out of 28 E11-HET mice developed spontaneous tumors (*Figure 4E*; *Supplementary file 1a–c*). Statistical analysis indicated that the tumor incidence was significantly higher for both *Trp73*[+/-] and E11-HET mice compared to that for WT mice (p=0.0211 for WT vs. *Trp73*[+/-]; p=0.0011 for WT vs. E11-HET by Fisher's exact test). Interestingly, although there was no difference in tumor incidence between E11-HET and *Trp73*[+/-] mice, the percentage of diffuse large B cell lymphoma (DLBCL) was significantly higher in E11-HET mice (50%) than that in *Trp73*[+/-] mice (14.8%) (*Figure 4E* and *Figure 4—figure supplement 1D*; p=0.009 by Fisher's exact test). These data suggest a role of p73γ in promoting B-cell lymphomagenesis, which is consistent with the observation that p73γ is frequently upregulated in dog lymphomas (*Figure 1E*).

In addition to tumors, E11-HET mice developed other pathological defects, including chronic inflammation, EMH, and hyperplasia in lymph node, thymus, or spleen. Specifically, 17 out of 27 *Trp73*[+/-] and 8 out of 28 E11-HET mice, whereas 0 out of 51 WT mice, showed chronic inflammation in three or more organs (*Figure 4F*, *Figure 4—figure supplement 1E*, and *Supplementary file 1a–c*). The chronic inflammation was much less severe in E11-HET mice compared to *Trp73*[+/-] mice (*Figure 4F*), suggesting that p73γ partially compensates p73α in suppressing inflammatory response. Moreover, compared to WT mice, *Trp73*[+/-] and E11-HET mice were prone to EMH (*Figure 4G*). Furthermore, *Trp73*[+/-] and E11-HET mice showed various degree of hyperplasia in immune organs, such as lymph node, spleen, and thymus, compared to WT mice (*Figure 4H–J*). Briefly, the percentage of splenic hyperplasia was higher in *Trp73*[+/-] and E11-HET mice than that in WT mice, with *Trp73*[+/-] even higher than E11-HET mice (*Figure 4H*). The percentage of lymphoid hyperplasia was higher in E11-HET mice

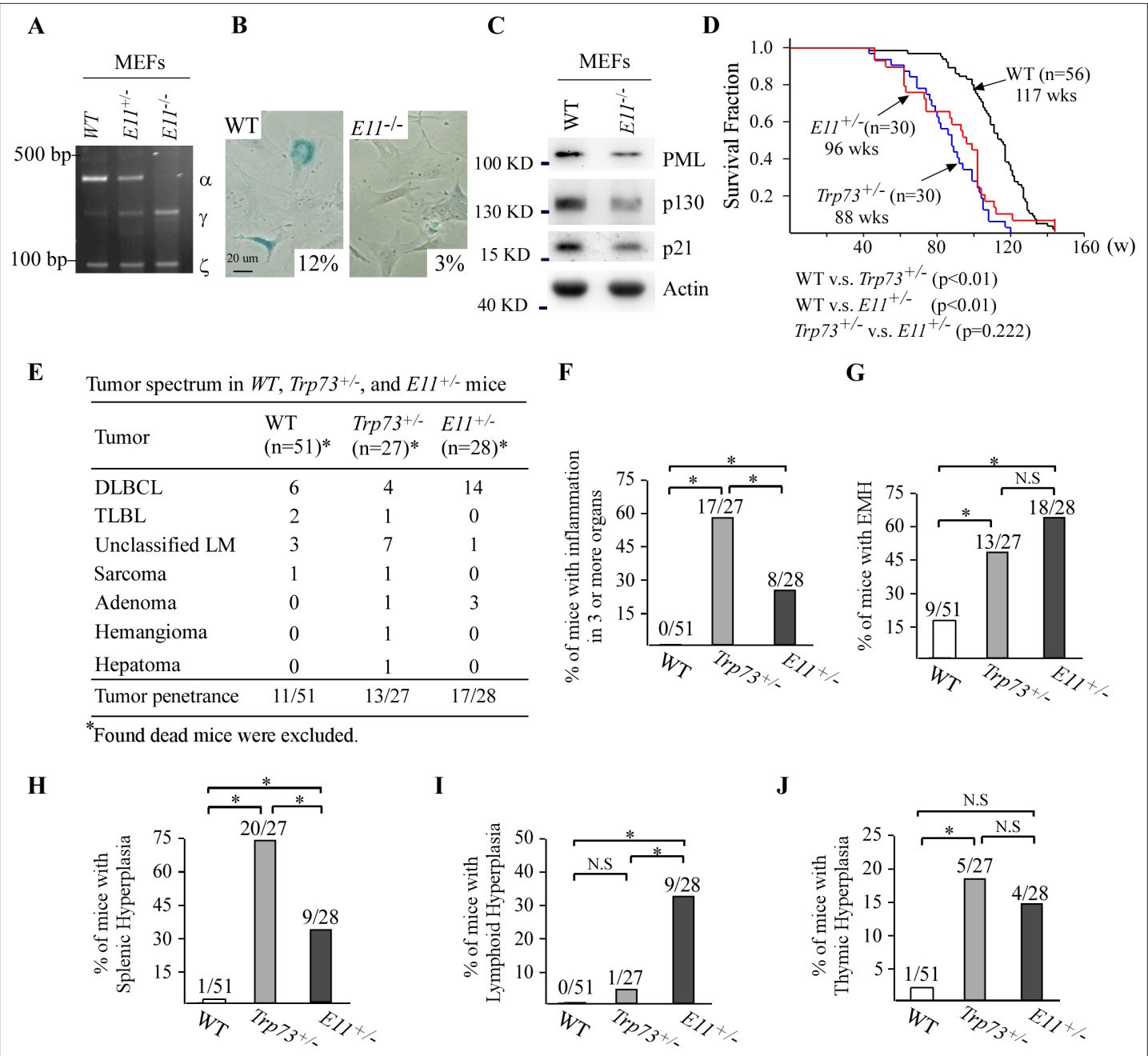

**Figure 4.** Deletion of E11 in the *Trp73* gene leads to shortened lifespan, increased incidence of spontaneous tumor and chronic inflammation in mice. (**A**) The levels of *Trp73α*, *Trp73γ*, *Trp73ζ* and *actin* transcripts were measured in WT, E11-HET, and E11-KO MEFs by RT-PCR analysis. (**B**) SA-β-gal staining was performed with WT and E11-KO MEFs. The percentage of SA-β-gal-positive cells was shown in each panel (WT vs. E11-KO: p<0.05 by Student's test).(**C**) Western blot was performed to measure the levels of PML, p130, p21, and actin proteins in WT and E11-KO MEFs using antibodies against PML, p130, p21, and actin. (**D**) Kaplan–Meier survival curves of WT (n = 56), *Trp73+/-* (n = 30), and E11-HET (n = 30) mice. (**E**) Tumor spectra in WT (n = 56), *Trp73+/-* (n = 27), and E11-HET (n = 28) mice. (**F**) Percentage of WT (n=51), *Trp73+/-* (n=27), and E11-HET (n=28) mice with inflammation in three or more organs.(**G**) Percentage of WT (n=51), *Trp73+/-* (n=27), and E11-HET (n=28) mice with extramedullary hematopoiesis. (**H**) Percentage of WT (n=51), *Trp73+/-* (n=27), and E11-HET (n=27) mice with splenic hyperplasia. (**I**) Percentage of WT (n=51), *Trp73+/-* (n=27), and E11-HET (n=28) mice with lymphoid follicular hyperplasia. (**J**) Percentage of WT (n=51), *Trp73+/-* (n=27), and E11-HET (n=28) mice with thymic hyperplasia.

The online version of this article includes the following source data and figure supplement(s) for figure 4:

**Source data 1.** Labeled gel images.

**Source data 2.** Raw gel images.

**Figure supplement 1.** Mice dificient in TP73 E11 were infertile and prone to diffuse large B cell lymphoma and chronic inflammation.

*Figure 4 continued on next page*

*Figure 4 continued*

**Figure supplement 1—source data 1.** Labeled gel images.

**Figure supplement 1—source data 2.** Raw gel images.

than that in WT and *Trp73*[+/-] mice (*Figure 4I*), whereas the percentage of thymic hyperplasia was higher in *Trp73*[+/-] mice than that in WT and E11-HET mice (*Figure 4J*). Together, these data indicated that E11-deficient mice phenocopies *Trp73*-deficient mice in short lifespan, infertility, chronic inflammation, and tumor incidence, indicating that p73γ cannot compensate p73α for tumor suppression and fertility.

## *E11*-deficient mice are prone to obesity

We noticed that E11-HET mice were bigger and fattier compared to WT mice. We also found that the size of adipocytes was larger in both male and female E11-HET mice than that in WT mice (*Figure 5A*). Likewise, a significant increase in visceral fat (VAT) mass was observed in both male and female E11-HET mice compared to WT mice (*Figure 5B annd C*). Moreover, compared to WT mice, E11-HET but not *Trp73*[+/-] mice showed increased incidence in liver steatosis as characterized by lipid deposition in the hepatocytes (*Figure 5D and E*). Consistently, the levels of transcripts for alanine aminotransferase (ALT), aspartate aminotransferase (AST), and γ-glutamyltransferase 1 (GGT1) (*Neuman et al., 2020*; *Sookoian et al., 2016*), all of which are associated with liver steatosis, were increased in E11-HET livers compared to WT liver (*Figure 5—figure supplement 1A*). These observations let us speculate that E11 deficiency promotes obesity in mice. To test this, the level of serum cholesterol and triglycerides was measured and found to be much higher in E11-HET mice than that in WT and *Trp73*[+/-] mice at ~65 wk (*Figure 5F and G*), suggesting that E11 deficiency enhances lipid storage. Consistent with this, younger E11-HET mice at 20 or 32 wk also showed increased body weights (*Figure 5H and I*) and elevated levels of serum cholesterol and triglycerides (*Figure 5J and K*) compared to WT mice. Furthermore, to examine whether p73γ is responsible for decreased lipid catabolism observed in E11-HET mice, the level of cholesterol and triglycerides was measured in isogenic control and E11-KO cells with or without knockdown of p73γ. We found that E11 deficiency led to marked increase in the level of cholesterol and triglycerides in both H1299 and Mia-PaCa2 cells (*Figure 5L–N* and *Figure 5—figure supplement 1B–D*). However, upon knockdown of *p73γ*, elevated level of cholesterol and triglycerides by E11-KO was reduced to normal level in isogenic control cells (*Figure 5L–N* and *Figure 5—figure supplement 1B–D*). By contrast, the level of cholesterol and triglycerides was increased by knockdown of *p73α/γ* in isogenic control H1299 and Mia-PaCa2 cells (*Figure 5L–N* and *Figure 5—figure supplement 1B–D*), suggesting a role of p73α in inhibiting lipid storage. Together, these data suggest that E11 deficiency leads to obesity in mice, which is largely induced by TAp73γ.

## E11 deficiency leads to elevated production of Leptin, a novel target of TAp73γ

Several studies have shown that dysfunctional adipose tissues secret adipocytokines, such as leptin, and subsequently promote adipose tissue and systemic inflammation, which is considered to be responsible for many obesity-related complications, including cancer (*Cao, 2014*; *Tilg and Moschen, 2006*; *Divella et al., 2016*). Interesting, we found that E11-HET but not WT mice showed pronounced low-grade inflammation in visceral adipose tissues (VATs) (*Figure 6—figure supplement 1A*) as well as increased expression of several proinflammatory cytokines, such as IL6, IL1α, and TNFα, in VAT and liver (*Figure 6A*). These observations prompted us to speculate whether E11 deficiency leads to altered expression of leptin, a pleiotropic adipocytokine that is known to be associated with obesity, adipose tissue inflammation, and cancer (*Myers et al., 2010*; *Abella et al., 2017*; *de Candia et al., 2021*). In addition to adipocytes, Leptin was found to be expressed in normal epithelial and carcinoma cells and subsequently promotes tumor growth through autocrine and paracrine signaling (*Lin and Hsiao, 2021*; *Kang and Moon, 2010*). Indeed, we found that the level of leptin transcript was increased by E11 deficiency in the VAT and liver tissues as well as in MEFs (*Figure 6A* and *Figure 6—figure supplement 1B*). Consistent with this, immunohistochemistry (IHC) assay showed an elevated expression of leptin protein in E11-HET kidney and liver compared to WT kidney and liver, respectively (*Figure 6—figure supplement 1C*). Moreover, the level of Leptin protein in the VAT and serum was

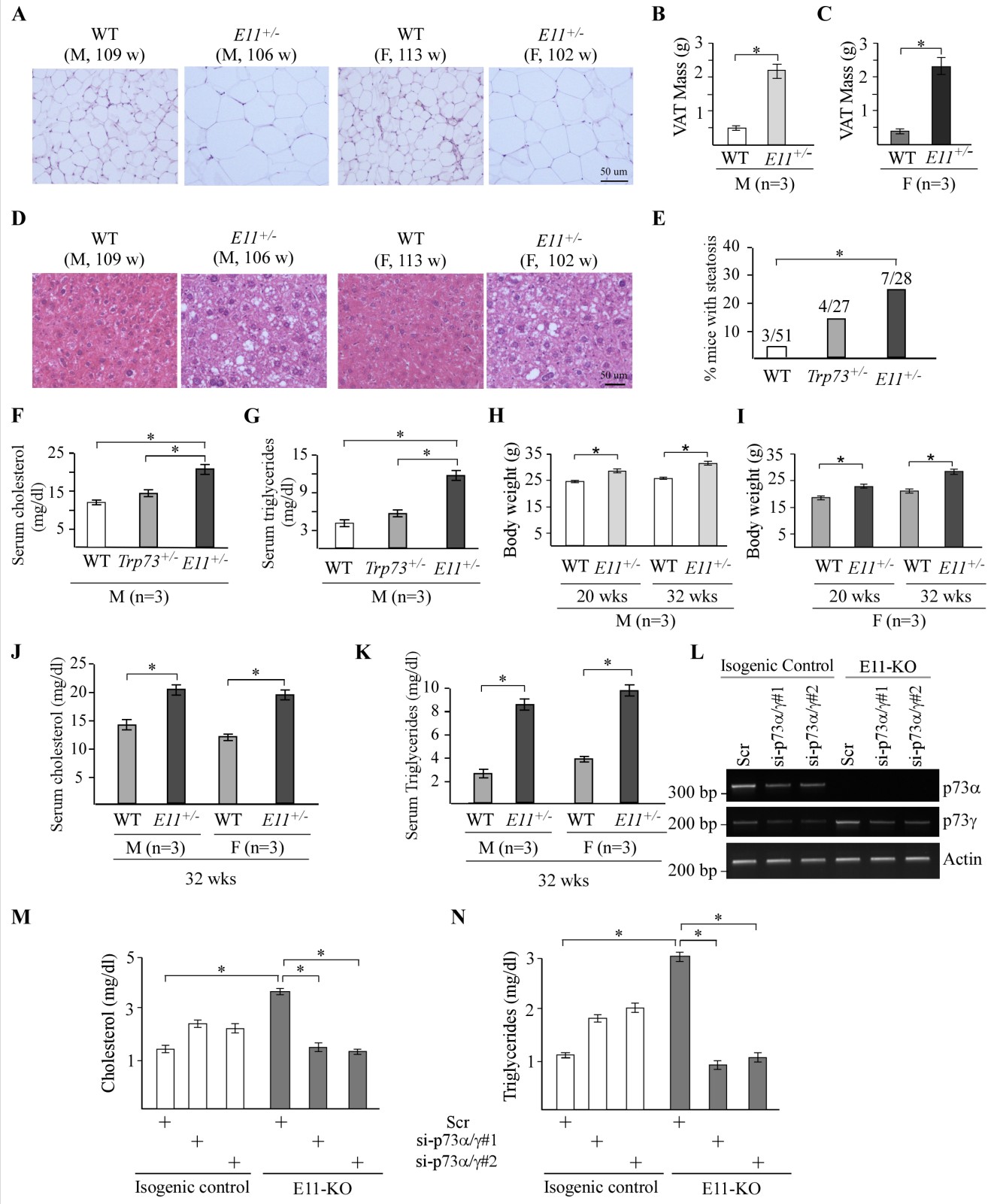

**Figure 5.** E11-deficient mice are prone to obesity. (**A**) Representative images of H&E-stained visceral adipose tissues from male and female WT and E11-HET mice. (**B–, C**) Visceral adipose tissue (VAT) mass of male (**B**) and female (**C**) WT and E11-HET mice at ~100 wk. * indicates p<0.05 by Fisher's exact test. (**D**) Representative images of H&E-stained liver tissues from male and female WT and E11-HET mice. (**E**) Percentage of WT (n=51), *Trp73*⁺/⁻ (n=27), and E11-HET (n=28) mice with liver steatosis. * indicates p<0.05 by Fisher's exact test. (**F, G**) The levels of serum cholesterol (**F**) and triglycerides

*Figure 5 continued*

(**G**) were measured in male WT, *Trp73*$^{+/-}$, and E11-HET mice at 100 wk. (**H, I**) Body weights were measured in male (**H**) and female (**I**) WT, *Trp73*$^{+/-}$, and E11-HET mice at 20 or 32 wk. (**J, K**) The levels of serum cholesterol (**J**) and triglycerides (**K**) were measured in male and female WT, *Trp73*$^{+/-}$, and E11-HET mice at 32 wk. (**L**) The levels of *p73α*, *p73γ*, and *actin* transcripts were measured in isogenic control and E11-KO H1299 cells transfected with a scrambled siRNA or siRNAs against p73α/γ. (**M, N**) Cells were treated as in (**L**), followed by measurement of cholesterol (**M**) and triglycerides (**N**). Data were presented as Mean ± SEM (n=3).

The online version of this article includes the following source data and figure supplement(s) for figure 5:

**Source data 1.** Labeled gel images.

**Source data 2.** Raw gel images.

**Source data 3.** Numerical data for *Figure 5*.

**Figure supplement 1.** p73gamma maintains the levels of triglycerides and cholesterol in cells.

**Figure supplement 1—source data 1.** Labeled gel images.

**Figure supplement 1—source data 2.** Raw gel images.

**Figure supplement 1—source data 3.** Numerical data for *Figure 5–figure supplement 1C–D*.

highly increased in age- and gender-matched E11-HET mice compared to that in WT and *Trp73*$^{+/-}$ mice (***Figure 6B and C***). While Leptin expression was increased in *Trp73*$^{+/-}$ mice it was still much lower than that in E11-HET mice (***Figure 6B and C***). Furthermore, the level of serum Leptin was found to be increased in both male and female E11-HET mice at very young ages of 20- or 32-week-old compared to WT mice (***Figure 6D and E***).

The observations above let us speculate whether increased expression of p73γ in E11-deficient mice contributes to Leptin expression. To test this, Leptin transcript was measured in VATs from age- and gender-matched WT, *Trp73*$^{+/-}$, and E11-HET mice. We found that the level of Leptin transcript was slightly increased in *Trp73*$^{+/-}$ mice, but markedly increased in E11-HET mice (***Figure 6F***), suggesting that Leptin expression is induced by p73γ. To verify this, Leptin transcript was measured in TP73-KO, TAp73-KO, ΔNp73-KO, and E11-KO H1299 cells. We found that knockout of all p73 or TAp73 decreased, whereas knockout of ΔNp73 had little effect on, the level of *Leptin* transcript (***Figure 6G***). However, the level of *Leptin* was much higher in E11-KO cells compared to isogenic control, TP73-, TAp73-, and ΔNp73-KO H1299 cells (***Figure 6G***). We also showed that ectopic expression of TAp73γ led to increased expression of Leptin (***Figure 6—figure supplement 1D***), whereas knockdown of p73γ markedly decreased the level of Leptin transcript in E11-KO H1299 and Mia-PaCa2 cells (***Figure 6H*** and ***Figure 6—figure supplement 1E***). Interestingly, knockdown of *p73α/γ* also led to reduction in Leptin expression in isogenic control cells in that p73α is the predominant isoform (***Figure 6H*** and ***Figure 6—figure supplement 1E***), suggesting that p73α can regulate Leptin transcription. To further test this, we found that ectopic expression of TAp73α, TAp73β, or TAp73γ alone was able to induce Leptin expression, with the strongest induction by TAp73γ (***Figure 6—figure supplement 1F***).

To determine whether Leptin is a direct target of TAp73, we searched the Leptin promoter and found a potential p73 response element (p73-RE) located at nt –346 to –321 (***Figure 6I***). Thus, ChIP assay was performed and showed that ectopic TAp73γ was able to bind to the LEP promoter (***Figure 6J***). As a positive control, TAp73γ bound to the p21 promoter (***Figure 6J***), a bona fide target of the p53 family (***El-Deiry et al., 1993***). Consistent with this, endogenous p73γ were able to bind to the Leptin promoter in E11-KO H1299 cells compared to isogenic control cells (***Figure 6K and L***). Furthermore, to determine whether the p73-RE (nt –346 to –321) is required for TAp73 to transactivate the Leptin promoter, luciferase reporters that contain WT or mutant p73-RE were generated (***Figure 6I***). We found that TAp73α/β/γ were able to activate the Leptin-WT reporter, with the strongest activation by TAp73γ (***Figure 6M***). By contrast, all TAp73 isoforms were unable to transactivate the luciferase reporter containing mutant p73-RE (***Figure 6M***). Together, these data indicate that Leptin is a novel target of TAp73γ and may play a role in TAp73γ-induced oncogenic activities.

## Leptin is a critical mediator of TAp73γ in oncogenesis and altered lipid metabolism

Leptin is known to play a critical role in oncogenesis through altered lipid metabolism (***Sánchez-Jiménez et al., 2019***; ***Sabol et al., 2019***; ***Park and Scherer, 2011***). Thus, we examined whether knockdown of *Leptin* has any effect on the level of cholesterol and triglycerides by using two different

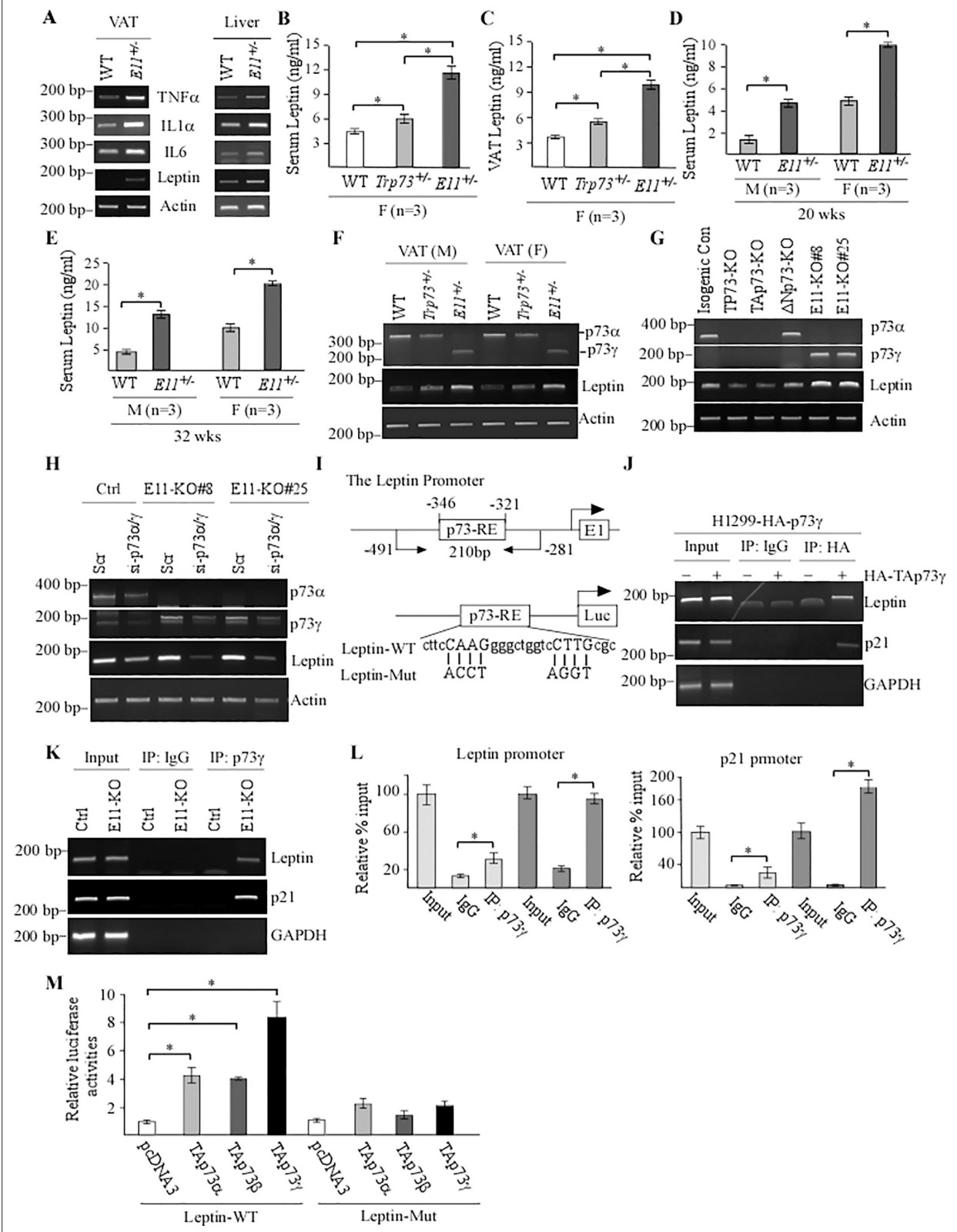

**Figure 6.** E11 deficiency leads to elevated production of Leptin, a novel target of TAp73g. (**A**) The levels of *TNFα*, *IL6*, *IL-1α*, *Leptin*, and *actin* transcripts were measured in visceral adipose tissue (VAT) and liver from WT and E11-HET mice. (**B, C**) The levels of serum (**B**) and VAT (**C**) Leptin were measured in female WT, *Trp73+/-*, and E11-HET mice at 65 wk. (**D, E**) The levels of serum leptin were measured in male and female WT and E11-HET mice at 20 (**D**) or 32 (**E**) wk. (**F**) The levels of *p73*, *Leptin*, and *actin* transcripts were measured in VATs from male and female WT, *Trp73+/-*, and E11-HET mice. (**G**) The

*Figure 6 continued on next page*

*Figure 6 continued*

levels of *p73α*, *p73γ*, *Leptin*, and *actin* transcripts were measured in isogenic control, total p73-KO, TAp73-KO, DNp73-KO, and E11-KO H1299 cells. (**H**) The levels of *p73α*, *p73γ*, *Leptin*, and *actin* transcripts were measured in isogenic control and E11-KO H1299 cells transfected with a scrambled siRNA or siRNA against for p73α/γ for 3 d. (**I**) Schematic representation of the Leptin promoter, the location of p73-RE as well as luciferase reporters with WT or mutant p73RE. (**J**) H1299 cells were uninduced or induced to express TAp73γ for 24 hr, followed by ChIP assay to measure the binding of TAp73γ to the Leptin, p21, and GAPDH promoter. (**K, L**) Isogenic control and E11-KO cells were used for ChIP assay to examine the binding of p73γ to Leptin and p21 promoter by regular PCR (**K**) or qPCR (**L**). Data were presented as Mean ± SEM (n=3).* indicates p<0.05 by Student's *t*-test. (**M**) Luciferase assay was performed with H1299 cells transfected with a control vector or vector expressing TAp73α, TAp73β, and TAp73g. The relative fold-change of luciferase activity was calculated as a ratio of luciferase activity of each construct versus an empty vector. Data were presented as Mean ± SEM (n=3). * indicates p<0.05 by Student's *t*-test.

The online version of this article includes the following source data and figure supplement(s) for figure 6:

**Source data 1.** Labeled gel images.

**Source data 2.** Raw gel images.

**Source data 3.** Numerical data for *Figure 6*.

**Figure supplement 1.** E11 dificiency alters lipid level via increased leptin levels.

**Figure supplement 1—source data 1.** Labeled gel images.

**Figure supplement 1—source data 2.** Raw gel images.

*Leptin* siRNAs in both H1299 and Mia-PaCa2 cells (*Figure 7A* and *Figure 7—figure supplement 1A*). We showed that knockdown of *Leptin* abrogated the elevated level of cholesterol and triglycerides by E11-KO in both H1299 and Mia-PaCa2 cells (*Figure 7B and C* and *Figure 7—figure supplement 1B and C*), consistent with the observation that knockdown of p73γ also abrogated the elevated level of cholesterol and triglycerides by E11-KO (*Figure 5M and N* and *Figure 5—figure supplement 1C and D*). Next, we examined the role of Leptin in cell migration and showed that in isogenic control cells, cell migration was moderately inhibited by knockdown of *Leptin* in both H1299 (*Figure 7D and E* and *Figure 7—figure supplement 1D*) and Mia-PaCa2 cells (*Figure 7—figure supplement 1E and F*). By contrast, the enhanced cell migration by E11-KO was abolished by knockdown of *Leptin* in both H1299 (*Figure 7D and E* and *Figure 7—figure supplement 1D*) and Mia-PaCa2 cells (*Figure 7—figure supplement 1E and F*), consistent with the observation that knockdown of *p73γ* also abolished the enhanced cell migration by E11-KO (*Figure 3C* and *Figure 7—figure supplement 1B*). To further analyze the role of Leptin in p73γ-mediated cell migration, we asked whether supplementation of Leptin would have an effect on cell migration. To make sure that the effect of Leptin is transmitted through its receptor, we measure the expression of Leptin receptor (LepR). We found that *LepR* was expressed at a similar level across isogenic control, TAp73-KO and E11-KO H1299 cells (*Figure 7—figure supplement 1G*), suggesting that these cells have an intact Leptin signaling pathway and that *LepR* expression is not regulated by p73. Importantly, we found that cell transmigration was enhanced by treatment with Leptin in isogenic control, TAp73-KO and E11-KO H1299 cells (*Figure 7—figure supplement 1H*). Furthermore, to determine whether the p73γ-Leptin axis plays a role in cancer progression, we examined whether the expression pattern of Leptin is correlated with that of p73γ in human prostate carcinomas and dog lymphomas. We found that Leptin expression was low in normal human prostates but highly elevated in prostate carcinomas, which correlates well with the expression of *p73γ* (*Figure 7F and G*, Leptin and p73γ panels, compare lanes 1–5 with 6–13). Similarly, *Leptin* and *p73γ* transcripts were found to be highly expressed coordinately in dog lymphomas compared to normal dog lymph nodes (*Figure 7H*, Leptin and p73γ panels, compare lanes 1–3 with 4–19). These data suggest that Leptin is a mediator of TAp73γ in oncogenesis and altered lipid metabolism and that the TAp73γ-Leptin pathway plays a role in the development of prostate cancer and lymphoma.

## Targeting p73γ or Leptin inhibits tumor growth in vivo

To explore the therapeutic potential of targeting the p73γ-Leptin pathway for cancer management, xenograft models were established by using isogenic control and E11-KO H1299 cells. We showed that loss of E11 led to enhanced tumor growth (*Figure 8A*). Consistent with this, tumor size and weight were significantly greater for E11-KO group than that for control group (*Figure 8B and C*). We also showed that the levels of *p73γ* and *Leptin* transcripts and proteins were much higher in E11-KO tumors than that in WT tumors (*Figure 8D* and *Figure 8—figure supplement 1A*). Additionally, H&E

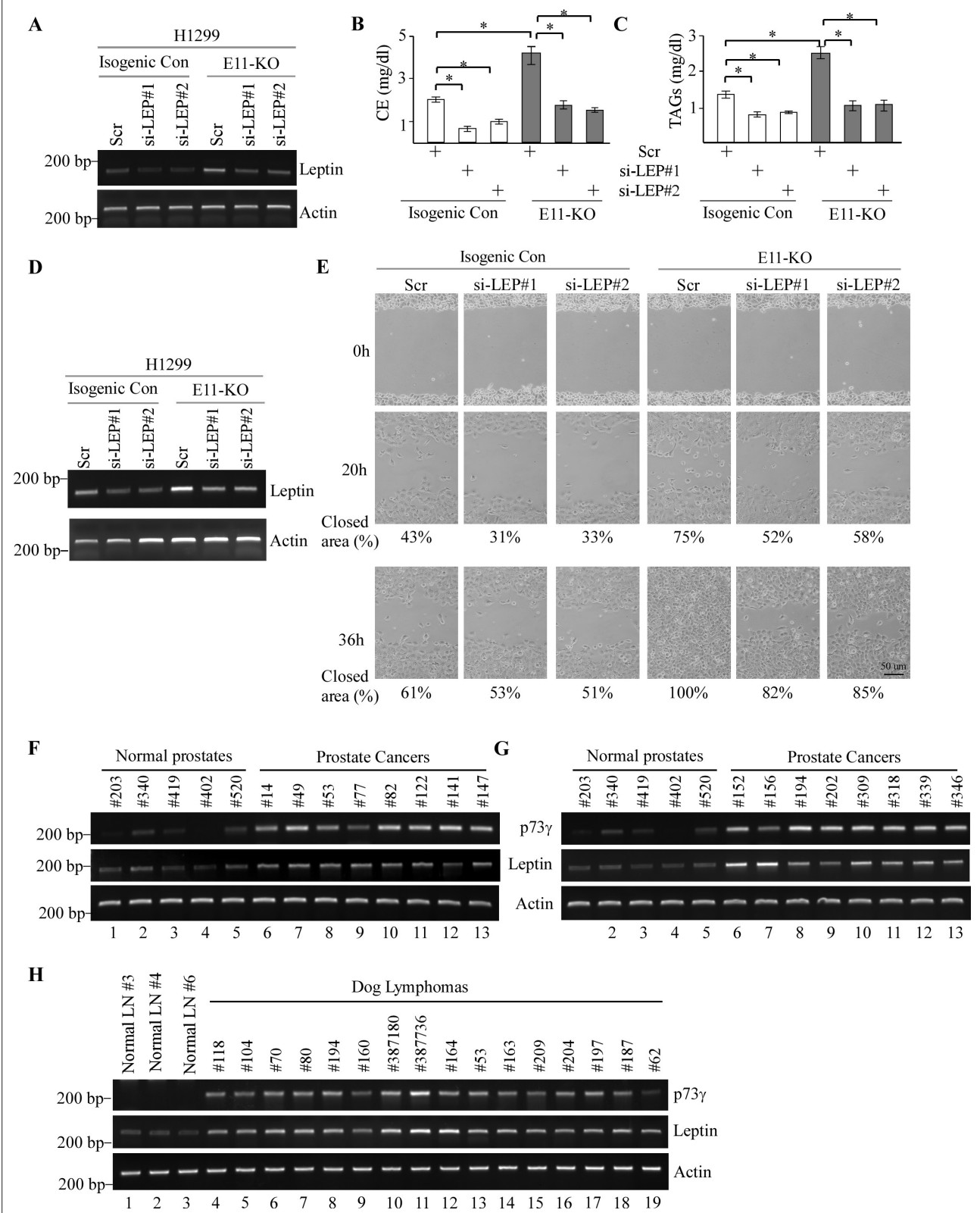

**Figure 7.** Leptin is a critical mediator of TAp73γ-mediated oncogenesis and altered lipid storage. (**A, D**) The levels of *Leptin* and *actin* transcripts were measured in isogenic control and E11-KO H1299 cells transfected with a scrambled siRNA or siRNAs against Leptin. (**B, C**) The levels of cholesterol (**B**) and triglycerides (**C**) were measured in cells treated as in (**A**). Data were presented as Mean ± SEM (n=3). * indicates p<0.05 by Student's *t*-test. (**E**) Scratch assay was performed with H1299 cells treated as in (**D**). Phase-contrast photomicrographs were taken immediately after scratch (0 hr), 20 hr, and

*Figure 7 continued on next page*

*Figure 7 continued*

36 hr later to monitor cell migration. (**F, G**) The levels of *p73γ*, *Leptin,* and *actin* transcripts were examined in 5 normal prostates and 16 human prostate carcinomas. (**H**) The levels of *p73γ*, *Leptin,* and *actin* were examined in 3 normal dog lymph nodes and 16 dog lymphomas.

The online version of this article includes the following source data and figure supplement(s) for figure 7:

**Source data 1.** Labeled gel images.

**Source data 2.** Raw gel images.

**Source data 3.** Numerical data for *Figure 7*.

**Figure supplement 1.** Leptin is required for p73gamma to enhance cell migration.

**Figure supplement 1—source data 1.** Labeled gel images.

**Figure supplement 1—source data 2.** Raw gel images.

analysis showed that both WT and E11-KO H1299 xenografts were human cancer cells (*Figure 8E*). Next, we determined whether the elevated levels of p73γ and Leptin are responsible for the enhanced tumor growth by E11-KO. To test this, siRNAs against *p73γ* or *Leptin* were synthesized along with a scrambled siRNA as a control. We found that the rate of growth for E11-KO tumors injected with *p73γ* or *Leptin* siRNA was much slower than the ones injected with the control scrambled siRNA (*Figure 8F*). Consistent with this, the tumor size and weight were much less for E11-KO tumors injected with *p73γ* or *Leptin* siRNA than the ones injected with the control scrambled siRNA (*Figure 8G and H*). As expected, we showed that both *p73γ* and *Leptin* were efficiently knocked down by their respective siRNAs (*Figure 8I* and *Figure 8—figure supplement 1B and C*). Notably, we found that treatment with *p73γ* or *Leptin* siRNA led to extensive tumor necrosis in xenografts compared with that treated with the control scrambled siRNA (*Figure 8J*). Together, we showed that E11-KO promotes, whereas knockdown of *p73γ* or *Leptin* suppresses, xenograft growth in mice.

## Discussion

The biology of p73 is complicated because p73 is expressed as multiple N- and C-terminal isoforms, some of which may have opposing functions. While TA and ΔNp73 isoforms are shown to exert opposing activities in tumorigenesis by multiple in vitro and in vivo model systems, the biological significance of C-terminal p73 splicing variants (α, β, and γ) remains largely uncharacterized. In this study, we aimed to manipulate alternative splicing of *TP73* using CRISPR-Cas9. We showed that deletion of E11 in *TP73* leads to isoform switch from p73α to p73γ in both human cancer cells and mice. Importantly, we found that owing to p73γ expression, E11 deficiency leads to enhanced tumorigenesis and obesity in vitro and in vitro. Moreover, we identified Leptin, a key hormone in energy homeostasis, as a novel target of p73 and a critical mediator of TAp73γ in oncogenesis and aberrant lipid metabolism. Furthermore, we showed that p73α-γ switch is detected in a subset of dog lymphomas, along with elevated expression of Leptin. Finally, to explore the therapeutic potential of the p73γ-Leptin pathway, we showed that E11-KO promotes, whereas knockdown of p73γ or Leptin suppresses, H1299 xenograft growth in mice. These results have provided compelling evidence and prompted us to hypothesize that p73 C-terminal isoforms, that is, TAp73α and TAp73γ, can exert two opposing functions in tumorigenesis in a manner similar to that by TA and ΔN p73 isoforms. These results may have also explained the discrepancies in several early studies that TAp73 have both pro- and anti-tumorigenic activities (*Dulloo et al., 2015b*; *Amelio et al., 2015*; *Li et al., 2018*; *Sharif et al., 2019*; *Sabapathy, 2015*; *Stantic et al., 2015*; *Dulloo et al., 2015a*; *Fernandez-Alonso et al., 2015*), which are exerted by TAp73α and TAp73γ, respectively. A model to elucidate the role of p73α/γ switch and the p73γ-Leptin axis in tumorigenesis is proposed and shown in *Figure 8K*.

We and others have shown that ectopic expression of TAp73α and TAp73γ can inhibit cell growth albeit to a much less extent compared to p53 or TAp73β (*Liu and Chen, 2005*; *Ueda et al., 1999*; *Nozell et al., 2003*; *Chi et al., 1999*), suggesting that TAp73α and TAp73γ have a role in tumor suppression. Here, we showed that when expressed at a physiological relevant level in vivo, TAp73γ possesses an oncogenic activity whereas TAp73α has a tumor-suppressive function. Specifically, we showed that knockout or knockdown of TAp73α, the major isoform in H1299 and Mia-PaCa2 cells, leads to enhanced cell proliferation and migration (*Figures 2E–G and 3C*, *Figure 2—figure supplement 1G–I*). By contrast, E11 deficiency, which switches TAp73α to TAp73γ, leads to enhanced cell

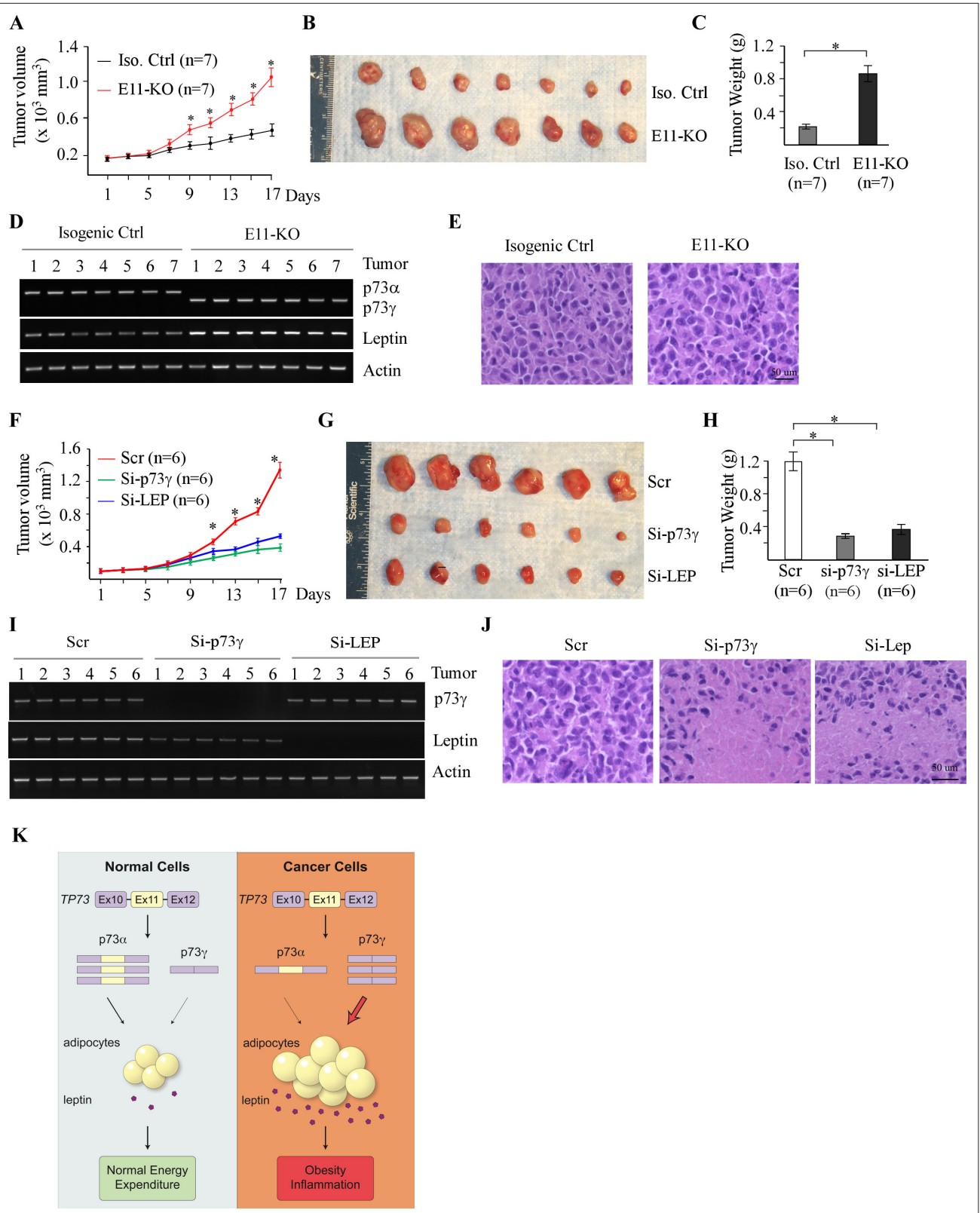

**Figure 8.** Targeting p73γ or Leptin inhibits tumor growth in vivo. (**A**) Xenograft tumor growth in nude mice from isogenic control or E11-KO H1299 cells (error bars represent SEM, * indicates p<0.05 by Student's t-test). (**B**) Images of tumors excised from isogenic control and E11-KO groups. (**C**) Tumor weight distribution between isogenic control and E11-KO groups upon termination of tumor growth experiments at day 17 (*p<0.05 by Student's t-test). (**D**) The levels of *p73α*, *p73γ*, *Leptin*, and *actin* transcripts were measured in the tumors from isogenic control and E11-KO groups. (**E**) Representative

*Figure 8 continued on next page*

*Figure 8 continued*

images of H&E-stained tumors from isogenic control and E11-KO groups. (**F**) Xenograft tumor growth in nude mice from E11-KO H1299 cells transfected with scrambled siRNA or a siRNA against Leptin or p73γ (error bars represent SEM, * indicates p<0.05 by Student's *t*-test). (**G**) Images of tumors excised from E11-KO groups with or without knockdown of Leptin or p73γ. (**H**) Tumor weight of E11-KO tumors with or without knockdown of Leptin or p73γ (*p<0.05 by Student's *t*-test). (**I**) The levels of *p73α*, *p73γ*, *Leptin*, and *actin* transcripts were measured in E11-KO tumors with or without knockdown of Leptin or p73γ. (**J**) Representative images of H&E-stained E11-KO tumors with or without knockdown of Leptin or p73γ. (**K**) A proposed model to elucidate the role of p73α/γ switch and the p73γ-Leptin axis in tumorigenesis.

The online version of this article includes the following source data and figure supplement(s) for figure 8:

**Source data 1.** Labeled gel images.

**Source data 2.** Raw gel images.

**Source data 3.** Numerical data for *Figure 8*.

**Figure supplement 1.** The levels of p73gamma and Leptin are measured in multiple sets of tumors.

**Figure supplement 1—source data 1.** Labeled gel images.

**Figure supplement 1—source data 2.** Raw gel images.

proliferation and migration as well as xenograft tumor growth, which can be attenuated by knock-down of p73γ (*Figures 2E–G, 3C and 8*, *Figure 2—figure supplement 1G–I*, *Figure 3—figure supplement 1B, C, F*). Moreover, mice deficient in *Trp73*, presumably p73α, were prone to sponta-neous tumors, such as lymphoma (*Figure 4E*), consistent with previous observations that loss of *Trp73* promotes Myc-induced B-cell lymphomas or mutant p53-mediated T-cell lymphomas (*Zhang et al., 2019*; *Nemajerova et al., 2010*). By contrast, E11-deficient mice were prone to spontaneous tumors, such as DLBCLs (*Figure 4E* and *Supplementary file 1c*), presumably due to elevated p73γ expression in these mice. Furthermore, we showed that while p73α was detectable in normal tissues, p73γ was elevated in a subset of dog lymphomas (*Figure 1C*). These data suggest that p73α and p73γ have opposing functions in tumorigenesis and that isoform switch from p73α to p73γ is a critical event for cancer progression. Indeed, we found that percentage of exon 11 exclusion, which would produce p73γ isoform, was much higher in tumor samples compared to normal tissues (*Figure 1A and B*). In line with this, TAp73γ transcript was found to be aberrantly expressed in human hematological malig-nancies, such as leukemia and non-Hodgkin's lymphomas (*Tschan et al., 2000*; *Alexandrova and Moll, 2012*; *Pluta et al., 2006*). Moreover, one study showed that in breast cancer p73α and p73β are the main isoforms expressed in normal breast tissues, whereas other C-terminal splice variants, including p73γ, are mainly detected in breast cancers (*Zaika et al., 1999*). However, the factor that contributes to the alternative splicing of p73 isoforms has not been identified, which worth further investigation. Indeed, several RNA-binding proteins, such as RBM38 and PCBP2 (*Ren et al., 2016*; *Yan et al., 2012*), are found to post-transcriptionally regulate p73 expression. Thus, it would be inter-esting to determine whether these RNA-binding proteins play a role in modulating alternative splicing of *TP73*. In addition, we noticed that surrounding sequence of *TP73* exon 11 acceptor site is different from that of exon 12. Thus, studies are needed to determine whether the acceptor site of exon 11 is a weak splicing site and thus, spliced out. Additionally, it is also important to identify a splicing factor that mediates exon 11 splicing out. We believe that addressing these questions will help better under-stand the role of p73γ in cancer, which would lay a foundation to develop a strategy to target p73γ for cancer management.

In addition to tumorigenesis, we noticed that p73α and p73γ have distinct and overlapped roles in multiple pathological processes based on the phenotypes from *Trp73*- and E11-deficient mice. We reasoned that if p73γ can compensate for loss of p73α, no abnormalities would be observed in E11-deficient mice. Instead, we found that both *Trp73*- and E11-deficient mice were runty, infertile, and prone to hydrocephalus (*Figure 4—figure supplement 1C* and data not shown; *Yang et al., 2000*). We also found that *Trp73*-deficient mice and to a lesser degree, E11-deficient mice, were susceptible to chronic inflammation and splenic hyperplasia (*Figure 4F and H*). These data indicate that p73α has distinct roles in hydrocephalus, fertility, chronic inflammation, and splenic hyperplasia, which cannot be fully substituted by p73γ. On the other hand, p73γ appeared to have a prominent role in lipid metabolism. Specifically, we found that E11-deficient mice were prone to spontaneous liver steatosis, increased body weight and VAT mass, and elevated serum triglycerides and cholesterol, all of which were not observed in *Trp73*-deficient mice (*Figure 5A–E*). To verify the role of p73γ in lipogenesis, we

showed that the elevated level of cholesterol and triglycerides by E11-KO was abrogated by knockdown of p73γ (*Figure 5L–N* and *Figure 5—figure supplement 1B–D*). Together, these data suggest that p73α and p73γ may have opposing roles in lipid metabolism in that p73γ promotes, whereas p73α inhibits, lipogenesis.

In this study, we found that Leptin expression is induced by p73γ, leading to elevated level of Leptin in serum, VAT, and possibly many other tissues (*Figure 6A* and *Figure 6—figure supplement 1B and C*). We also found that accumulation of cholesterol/triglycerides and enhanced cell migration by E11 deficiency is abrogated by knockdown of Leptin (*Figure 7B, C, and E*; *Figure 7—figure supplement 1B–F*), which phenocopies the effect of p73γ-knockdown in E11-KO cells (*Figure 3C* and *Figure 2—figure supplement 1G–I* and *Figure 5—figure supplement 1C and D*). In line with this, we found that Leptin treatment was able to promote cell transmigration in isogenic control, TAp73-KO and E11-KO H1299 cells (*Figure 7—figure supplement 1H*), indicating that Leptin functions as a mediator of p73 in promoting cell migration. To uncover the mechanism, we showed that Leptin is a target of p73γ (*Figure 6F–L*). Interestingly, we noticed that while capable, p73α is much weaker than p73γ to induce Leptin expression (*Figure 6L* and *Figure 6—figure supplement 1F*). One possibility is that the unique C-terminal domains in p73α and p73γ can recruit distinct sets of transcription coactivators and, thus, differentially regulate gene expression, including Leptin, which warrants further investigation.

To explore the physiological significance of the p73γ-Leptin pathway, we found that Leptin expression correlates well with isoform switch from p73α to p73γ in a set of dog lymphomas (*Figure 7F*). Leptin was initially found to control energy homeostasis through the hypothalamus and was given a high hope for treating obesity (*Friedman, 2010*). However, it was found later that both too little and too much Leptin can lead to obesity (*Considine et al., 1996*; *Considine and Caro, 1996*), referred to as 'leptin resistance' (*El-Haschimi et al., 2000*; *Pelleymounter et al., 1995*). At this moment, it is uncertain whether Leptin directly contributes to obesity mediated by p73γ, partially due to the above-mentioned paradoxical roles of Leptin in obesity. However, since obesity is now considered a risk factor for many cancers, the p73γ-Leptin pathway may provide a physiological link between cancer and obesity. Indeed, it should be noted that many studies, including data from this study, indicate that Leptin and its receptor are expressed in both normal and tumor tissues and that Leptin is found to promote inflammation, tumor growth, and angiogenesis (*Sánchez-Jiménez et al., 2019*; *Dutta et al., 2012*). Considering that the p73γ is highly expressed in tumors such as dog lymphoma (*Figure 1C*), it is possible that p73γ plays a critical role in activating the Leptin signaling pathway in tumors, which subsequently promotes tumorigenesis. Moreover, we found that knockdown of p73γ or Leptin was able to inhibit growth of E11-KO xenograft tumors (*Figure 8*), which would open a new revenue for cancer management by targeting the p73γ-Leptin pathway. Together, these data indicate that as a target of p73, Leptin mediates p73γ in tumor promotion and altered lipid metabolism and that the p73γ-Leptin pathway can be targeted for cancer management.

# Materials and methods

## Reagents

Anti-Actin (sc-47778, 1:3000), anti-p130 (sc-374521, 1:3000), anti-p21 (sc-53870, 1:3000), anti- p130 (sc-374521, 1:3000), and anti-PML (sc-377390, 1:3000) were purchased from Santa Cruz Biotechnology. Anti-TAp73 (A300-126A, 1:2000) was purchased from Bethyl Laboratories. Anti-HA (MMS-101P, 1:3000) was purchased from Covance. Anti-Ki-67 (Cat# 12202, 1:100), anti-B220 (Cat# 70265, 1:100), and anti-Leptin (Cat# 16227, 1:500) were purchased from Cell Signaling. Anti-p73γ/ε antibody was generated by Cocalico Biologicals using a peptide (PRDAQQPWPRSASQRRDEC). To-Pro-3 was purchased from Invitrogen. The WesternBright ECL HRP substrate (Cat# K12043-D20) was purchased from Advansta. Matrigel was obtained from Corning Inc (Cat# 354230). IHC kit (Cat# PK6100) and DAB (Cat# SK4100) were purchased from Vector Laboratories. Dual-Luciferase Reporter Assay kit was purchased from Promega (Cat# E1910). Scrambled siRNA (5'-GGC CGA UUG UCA AAU AAU U-3'), p73α/γ siRNA#1 (5'-AGC CUC GUC AGU UUU UUA A-3'), p73α/γ siRNA#2 (5'-AAC CUG ACC AUU GAG GAC CUG GG-3'), Leptin siRNA#1 (5'-GCU GGA AGC ACA UGU UUA U-3'), and Leptin siRNA#2 (5'-CCA GAA ACG UGA UCC AAA UUU-3') were purchased from Dharmacon (Chicago, IL). For siRNA transfection, RNAiMax (Life Technologies) was used according to the user's manual. Proteinase inhibitor cocktail was purchased from Sigma-Aldrich. Magnetic Protein A/G beads were purchased from

MedChem. RiboLock RNase Inhibitor and Revert Aid First Strand cDNA Synthesis Kit were purchased from Thermo Fisher. Recombinant Human Leptin Protein was purchased from R&D (398-LP).

## Plasmids

To generate a vector expressing a single-guide RNA (sgRNA) that targets total p73, TAp73, ΔNp73, and E11, two 25-nt oligos were annealed and then cloned into the pSpCas9(BB) sgRNA expression vector via BbsI site (*Ran et al., 2013*). To generate the pSpCas9(BB)-ΔNp73-sgRNA-Blast vector expressing both blasticidin and ΔNp73 sgRNA, the blasticidin gene was amplified from pcDNA6 vector and then cloned into pSpCas9(BB)-ΔNp73 via NotI. To generate a luciferase reporter that contains the WT Leptin promoter, a DNA fragment was amplified from genomic DNA from H1299 cells, cloned into pGL2-basic vector via XmaI and HindIII sites, followed by sequence confirmation. To generate a luciferase reporter that contains mutant Leptin promoter, two-step PCR was used. The first step was to amplify two DNA fragments (fragment #1 and #2) using the WT Lep promoter as the template. Fragment #1 was amplified using WT forward 1 primer and mutant reverse mutant 1 primer. Fragment #2 was amplified using mutant forward 1 primer and WT reverse 1 primer. The second round of PCR was performed using both fragment #1 and #2 as templates with a WT forward 1 and a reverse 1 primer. The resulted DNA fragment was then cloned into pGL2-basic vector via XmaI and HindIII sites. To generate a pcDNA4 vector expressing HA-tagged TAp73γ, a 680 bp DNA fragment was amplified using cDNAs from H1299 cells as a template, and then used to replace the C-terminal of HA-TAp73α (*Liu and Chen, 2005*) via EcoRI and XhoI sites. The sequences of all primers are listed in *Supplementary file 2a*.

## Mice and MEF isolation

*Trp73^{+/−}* mice were generated as described previously (*Zhang et al., 2019*). E11-HET mice were generated by the Mouse Biology Program at University of California at Davis. All animals and use protocols were approved by the University of California at Davis Institutional Animal Care and Use Committee. To generate WT and E11-KO MEFs, E11-HET mice were bred with E11-HET and MEFs were isolated from 12.5 to 13.5 postcoitum (p.c.) mouse embryos as described previously (*Zhang et al., 2011a*). MEFs were cultured in DMEM supplemented with 10% FBS (Life Science Technology), 55 µM β-mercaptoethanol, and 1× non-essential amino acids (NEAA) solution (Cellgro). All the genotyping primers are listed in *Supplementary file 2b*.

## Cell culture, cell line generation

Cells from this study were obtained from the ATCC between 2003 and 2016 and used below passage 25 and tested negative for mycoplasma after thawing. Because all cell lines from the ATCC have been thoroughly tested and authenticated, we did not authenticate the cell lines used in this study. H1299 and Mia-PaCa2 cells and their derivatives were cultured in Dulbecco's modified Eagle's medium (DMEM, Invitrogen) supplemented with 10% fetal bovine serum (Hyclone). MCF-10A cells were cultured in DMEM:F12 (1:1) supplemented with 5% donor horse serum, 20 ng/ml EGF, 10 µg/ml insulin, 0.5 µg/ml hydrocortisone, and 100 ng/ml cholera toxin. To generate Total p73-, TAp73, ΔNP73, and E11-KO cell lines by CRISPR-Cas9, H1299 or Mia-PaCa2 cells were transfected with pSpCas9(BB)–2A-Puro vector expressing a guide RNA, and then selected with puromycin for 2–3 wk. To generate ΔN/E11-DKO cells, E11-KO H1299 and Mia-PaCa2 cells were transfected with pSpCas9(BB)-ΔNp73-sgRNA-Blast vector, followed by selection with blasticidin for 2–3 wk. Individual clone was picked confirmed by sequence analysis or western blot analysis. The primers used for sequencing total p73, TAp73, ΔNp73, and p73 exon 11 are listed in *Supplementary file 2b*. H1299 cells that inducibly express HA-tagged TAp73γ were generated as previously described (*Dohn et al., 2003*). Briefly, pcDNA4-HA-TAp73 vector was transfected into H1299 cells in which a tetracycline repressor is expressed by pcDNA6 (*Yan and Chen, 2006*). Individual clone was picked and screened and for TAp73γ expression by performing western blot analysis. To induce HA-tagged TAp73γ expression, tetracycline was added to the medium.

## Western blot analysis

Western blot analysis was performed as previously described (*Dohn et al., 2001*). Briefly, whole-cell lysates were harvested by 2× SDS sample buffer. Proteins were separated in 7–13% SDS-polyacrylamide gel, transferred to a nitrocellulose membrane, probed with indicated antibodies,

followed by detection with enhanced chemiluminescence and visualized by VisionWorksLS software (Analytik Jena).

## RNA isolation, RT-PCR, and qRT-PCR

Total RNA was isolated with Trizol reagent as described according to the user's manual. cDNA was synthesized with Reverse Transcriptase (Promega, San Luis Obispo, CA) and used for RT-PCR. The PCR program used for amplification was (1) 94°C for 5 min, (2) 94°C for 45 s, (3) 58°C for 45 s, (4) 72°C for 30 s, and (5) 72°C for 10 min. From steps 2–4, the cycle was repeated 22 times for actin and GAPDH, 28–35 times depending on the targets. All primers used for RT-PCR are listed in *Supplementary file 2c*. For qPCR, PowerUp Syber Green Master Mix (Applied Biosystems, Cat# A25742) was used according to the manufacturer's protocol.

## ChIP assay

ChIP assay was performed as previously described (*Harms and Chen, 2005*). Briefly, chromatin was cross-linked in 1% formaldehyde in phosphate-buffered saline (PBS). Chromatin lysates were sonicated to yield 200- to 1000-bp DNA fragments and immunoprecipitated with a control IgG or an antibody against HA or p73γ. After reverse cross-linking and phenol-chloroform extraction, DNA fragments were purified, followed by PCR to visualize the enriched DNA fragments. The primers used for the ChIP assays are listed in *Supplementary file 2c*.

## Luciferase reporter assay

Dual luciferase assay was performed according to the manufacturer's instructions (Promega). Briefly, H1299 cells were plated at $5 \times 10^4$ cells in triplicate per well in a 24-well plate and allowed to recover overnight. Cells were then transfected with the following plasmids: (1) 3 ng of pRL-SV40-Renilla; (2) 0.25 ug of a luciferase reporter, and (3) 0.25 ug of empty pcDNA3 vector or a pcDNA3 vector expressing p73α, β, or γ. The relative fold change of luciferase activity is a product of the luciferase activity induced by p73α, β, or γ divided by that induced by control vector.

## Colony formation assay

For colony formation assay, cells (600 per well) in a 6-well plate were cultured for 15 d. The cell clones were fixed with methanol/glacial acetic acid (7:1) and then stained with 0.1% of crystal violet.

## Wound-healing assay

$2 \times 10^5$ cells were seeded in a 6-well plate cells and grown for 24 hr. The monolayers were wounded by scraping with a P200 micropipette tip and washed two times with PBS. At indicated time points after scraping, cell monolayers were photographed with phase-contrast microscopy. Cell migration was determined by visual assessment of cells migrating into the wound. Wound closure percentage was quantified using ImageJ plugin, Wound Healing Sizing Tool (*Suarez-Arnedo et al., 2020*), by comparing the width of the wound between 0 hr and indicated time points.

## Senescence assay

The senescence assay was performed as described previously (*Qian et al., 2008*). Briefly, primary MEFs at passage 5 were seeded at $5 \times 10^4$ in a well of 6-well plate for 24 hr. Cells were then washed with 1× PBS and fixed with 2% formaldehyde, 0.2% glutaraldehyde for 15 min at room temperature, and then stained with fresh SA-$\beta$-galactosidase staining solution (1 mg/ml 5-bromo-4-chloro-3-indolyl-β-d-galactopyranoside, 40 mm citric acid/sodium phosphate [pH 6.0], 5 mm potassium ferrocyanide, 5 mm potassium ferricyanide, 150 mm NaCl, and 2 mm $MgCl_2$). The percentage of senescent cells was calculated as the number of SA-β-gal-positive cells divided by the number of total cells counted.

## Histological analysis and IHC

Mouse tissues were fixed in 10% (wt/vol) neutral-buffered formalin, processed, and embedded in paraffin blocks. Tissues blocks were sectioned (6 μm) and stained with hematoxylin and eosin (H&E). IHC analysis was performed using the Vectastain ABC Elite Kit (Vector Laboratories) according to the manufacturer's instruction. Briefly, tissue sections (5 μm) were dewaxed and antigen-retrieved in a citrate buffer (pH 6.0), followed by incubation with a primary antibody anti-Ki-67 (Cat# 12202,

1:100), anti-B220 (Cat# 70265, 1:100), or anti-Leptin (Cat# 16227, 1:500) overnight at 4°C and then a secondary antibody for 1 hr at room temperature. The slides were visualized by treatment with 3,3'-diaminobenzidine tetrahydrochloride (DAB), and then counterstained with Mayer's hematoxylin.

## ELISA

Leptin ELISA was performed using the BioVendor Mouse and Rat Leptin Elisa kit (Cat# RD291001200R). Briefly, mouse serum was incubated with a microplate precoated with mouse leptin antibody. Bound leptin was detected with biotin-labeled polyclonal anti-mouse leptin antibody conjugated to horse-radish peroxidase and quantified by a chromogenic substrate at 450 nm. A standard curve was constructed by plotting absorbance values versus leptin concentrations of standards, and concentrations of unknown samples were determined using this standard curve. To measure the level of cholesterol and triglycerides, Cholesterol/Cholesterol Ester-Glo Assay kit (Cat# J3190, Promega) and Triglyceride-Glo Assay kit (Cat# J3160, Promega) from Promega were used according to the user's manual. Briefly, mouse sera or cell lysates were first incubated with cholesterol/triglycerides lysis solution at 37°C for 30 min, then incubated with cholesterol/triglycerides detection reagent for 1 hr, followed by luminescence detection with Luminometer (SpectraMAX). A standard curve was constructed and concentrations of unknown samples were determined using the standard curve.

## Three-dimensional culture for acini

The assay was performed as previously described (*Zhang et al., 2011b*). Briefly, single-cell suspensions were plated onto Matrigel-coated chamber slides at 5000 cells/well in complete growth medium with 2% Matrigel and allowed to grow for 1–22 d. Overlay medium containing 2% Matrigel was renewed every 4 d. At the end of culture, cells were fixed and the nuclei were stained with 5 μg/ml of To-Pro-3 in PBS for 15 min at room temperature. Confocal microscopic images of the acinus structures were captured by the Z-stacking function for serial confocal sectioning at 2 μm intervals (LSM-510 Carl Zeiss laser scanning microscope) and then analyzed using Carl Zeiss software.

## Three-dimensional tumor spheroid culture

Single-cell suspensions (3000 cells/well) were plated around the rim of the well of a 96-well plate in a 4:3 mixture of Matrigel (BD Biosciences CB-40324)and MammoCult medium (Stemcell technology, Catalog # 05620). The cell mixture was then incubated at 37°C with 5% $CO_2$ for 15 min to solidify the gel, followed by addition of 100 μl of pre-warmed MammoCult. At the end of 3-D culture, cells were released from the Matrigel by incubating with 50 μl of dispase (5 mg/ml) (Life Technologies #17105-041) at 37°C for 45 min. Spheroids were imaged using a phase-contrast microscope and cell viability was measured by CellTiter-Glo according to the manufacturer's guidelines (Promega, Cat#G9681).

## Transwell migration assay

Transwell migration assay was performed as previously described (*Justus et al., 2014*). Briefly, 1 × $10^5$ cells were seeded in 100 μl of serum-free medium in the upper chamber of a 24-well transwell and then incubated at 37°C for 10 min to allow cells to settle down. Next, 600 μl of the DMEM with 10% FBS were added to the lower chamber in a 24-well transwell and cells were cultured at 37°C for various time. At the end of each time point, the transwell insert was removed from the plate and cells that had not migrated were removed with a cotton-tipped applicator. Cells that migrated to the other side of membrane were fixed with 70% ethanol for 20 min, stained with 0.1% of crystal violet, and photographed by phase-contrast microscope. For siRNA knockdown experiment, isogenic control and E11-KO H1299 cells were transfected with a scrambled siRNA or an siRNA against p73α/γ or Leptin for 48 hr, and then subjected to transwell assays. For Leptin treatment, recombinant Leptin protein (100 ng/ml) was added to media for 10 hr.

## Tissue collection

The human normal and prostate cancer specimens were obtained from Dr. Ralph De Vere White's group, with consent from patients who underwent radical prostatectomie. Fresh frozen dog lymph nodes from clinical samples were provided by the University of California at Davis Small Animal Clinic with the owner's permission. Samples were homogenized in Trizol, followed by RNA and protein purification according to the user's manual.

## Xenograft assay

$6 \times 10^6$ cells were mixed with Matrigel (1:1 ratio) in 100 ul and then injected subcutaneously into 8-week-old BALB/c athymic nude mice (Charles River). When tumors were palpable, tumor growth was monitored for every 2 d for a period of up to 17 d. To knock down p73γ and Leptin in vivo, Accell siRNAs against p73γ or Leptin were synthesized from Dharmacon, which were then transiently transfected into E11-KO H1299 cells along with a scrambled Accell siRNA at a concentration of 7.5 μM. Two days post transfection, cells were collected and injected into athymic nude mice, followed by monitoring tumor growth as describe above. Mice bearing xenograft tumors also received one more intratumoral injection of Accell siRNAs (7.5 μM) at day 9. Tumor volume was calculated according to the standard formula: $V$ = length × width × depth × 0.5236 (*Janik et al., 1975*). At the endpoint, all animals were sacrificed and the tumors weighed. One half of the tumor was stored at −80°C and the other half fixed in formalin and embedded with paraffin. Tumors were sectioned and H&E-stained for histopathology examination. All animals and use protocols were approved by the University of California at Davis Institutional Animal Care and Use Committee.

## Statistical analysis

The log-rank test was used for Kaplan–Meier survival analysis. Fisher's exact test or two-tailed Student's *t*-test was performed for the statistical analysis as indicated. $p < 0.05$ was considered significant.

## Acknowledgements

This work was supported in part by the National Institutes of Health R01 grants (CA081237 and CA224433) and UC Davis Cancer Center Core Support Grant CA093373 to X Chen and by Tobacco-related disease research program (T31IP1727) and the CCAH (Center for Companion Animal Health, UC Davis) grants 2019-13F and 2021-7F to J Zhang.

---

## Additional information

### Competing interests

Hee Jung Yang: is affiliated with LG Chem Ltd. The author has no financial interests to declare. The other authors declare that no competing interests exist.

### Funding

| Funder | Grant reference number | Author |
|---|---|---|
| National Institutes of Health | CA081237 | Xinbin Chen |
| National Institutes of Health | CA224433 | Xinbin Chen |
| National Institutes of Health | CA093373 | Xinbin Chen |
| Tobacco-Related Disease Research Program | T31IP1727 | Jin Zhang |
| Center for Companion Animal Health, University of California, Davis | 2019-13-F | Jin Zhang |
| Center for Companion Animal Health, University of California, Davis | 2021-7-F | Jin Zhang |

The funders had no role in study design, data collection and interpretation, or the decision to submit the work for publication.

---

## Author contributions

Xiangmudong Kong, Data curation, Formal analysis, Methodology, Writing – original draft; Wensheng Yan, Wenqiang Sun, Yanhong Zhang, Hee Jung Yang, Data curation, Formal analysis, Methodology; Mingyi Chen, Formal analysis; Hongwu Chen, Resources, Data curation, Formal analysis; Ralph W de Vere White, Resources; Jin Zhang, Conceptualization, Data curation, Formal analysis, Funding acquisition, Methodology, Writing – original draft, Project administration, Writing - review and editing; Xinbin Chen, Conceptualization, Data curation, Formal analysis, Supervision, Funding acquisition, Methodology, Writing – original draft, Project administration, Writing - review and editing

## Author ORCIDs

Xiangmudong Kong (ID) https://orcid.org/0000-0002-6835-920X
Xinbin Chen (ID) https://orcid.org/0000-0002-4582-6506

## Ethics

All animals and use protocols were approved by the University of California at Davis Institutional Animal Care and Use Committee.

## Decision letter and Author response

Decision letter https://doi.org/10.7554/eLife.82115.sa1
Author response https://doi.org/10.7554/eLife.82115.sa2

## Additional files

### Supplementary files

Supplementary file 1. Survival time, tumor spectrum, steatosis, inflammation, and other abnormalities in WT, *Trp73*[+/-] and E11[+/-] mice. (**a**) Wild-type (WT) mice (n = 56) – survival time, tumor spectrum, steatosis, inflammation, and other abnormalities. (**b**) *Trp73*[+/-] mice (n = 30) – survival time, tumor spectrum, steatosis, inflammation, and other abnormalities. (**c**) E11-HET mice (n = 30) – survival time, tumor spectrum, steatosis, inflammation, and other abnormalities.

Supplementary file 2. Primers for generating vectors. (**a**) The primers used to generate plasmids' expression vectors. (**b**) The primers used for genotyping. (**c**) The primers used for RT-PCR and ChIP.

MDAR checklist

### Data availability

The authors confirm that the data supporting the findings of this study are available within the article, the source data, and its supplementary materials.

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
