## [Editor Report]

TP73 is a member of the p53 family of tumor suppressors and is expressed as TAp73 and DNp73 and multiple C-terminal isoforms as a result of alternative splicing. This manuscript describes an interesting study revealing the complex and intricate functional network driven by p73 isoforms. Using elegant in vitro and in vivo assays the authors provide compelling evidence that a TAp73-α to TAp73-γ switch could be a frequent phenomenon in human cancers and provide novel evidence that TAp73-γ has oncogenic functions via Leptin.

---

## [Decision Letter]

**Decision letter after peer review:**

Thank you for submitting your article "Isoform-specific Disruption of the TP73 Gene Reveals a Critical Role for TAp73gamma in Tumorigenesis via Leptin" for consideration by *eLife*. Your article has been reviewed by 3 peer reviewers, one of whom is a member of our Board of Reviewing Editors, and the evaluation has been overseen by Richard White as the Senior Editor. The reviewers have opted to remain anonymous.

Essential revisions:

TP73 is a member of the p53 family of tumor suppressors and is expressed as TAp73 and DNp73 and multiple C-terminal isoforms as a result of alternative splicing. In this study, the authors used isoform-specific disruption of the TP73 gene to investigate the physiological functions of p53 C-terminal isoforms, focussing on p73a and p73g. They identify an oncogenic role of TAp73-γ in tumorigenesis via regulating the expression of a novel target Leptin. Furthermore, they generated and characterized a mouse mode that expresses the TAp73 isoform γ but not α and shows how this splicing switch has oncogenic effects and causes metabolic defects. Overall, this is an important and well-done study uncovering a key role of TAp63-g in tumorigenesis via regulating Leptin expression.

1. It appears unclear why the authors focus their analysis on a relatively small number of non-tumoral and tumoral prostate tissues. Did they analyze matched non-tumoral vs tumoral tissues? A table resuming the clinical features of the analyzed patients' needs to be included. Do the anyalzed samples present loss of exon 11?

2. Interestingly, the authors reported that "E11+/- mice were bigger and fattier as compared to WT mice" with evident signs of liver steatosis. These findings might be, at least in part, mechanistically supported by TAp73γ-induced Leptin expression. A number of experiments in vitro and in vivo have been elegantly performed to show altered expression of the axis TAp73γ/Leptin in H1299 and MiaPaCa2 engineered cell lines. While these findings can be mechanistically provocative there is no any link with NSCL and prostate cancers in humans. Is Leptin expression altered in NSCL and prostate cancer databases? This is critical to translate the reported findings into a realistic cancer context. An experimental effort on this direction should be included.

3. Statistical analysis is lacking throughout the manuscript. Figures 6L, 7B, 7C and supplemental Figure 5C and D do not have any statistics. The rest of the figures have asterisks but what do they mean? The asterisks are not explained and there is no footnote for p-values. Also, it is not clear sometimes how many replicates each experiment was conducted especially for migration and scratch would assays.

4. It is not very clear why the authors chose to do semi quantitative PCR and not RT-qPCR throughput the manuscript. Given that many of the changes are quite modest (example Figure 3A), RT-qPCR should be performed for the siRNA experiments. Likewise, for the ChIP experiments in Figure 6J, ChIP-qPCR should also be performed to get a sense of the magnitude of the enrichment.

5. What are the global transcriptional differences between TAp73 α and γ isoforms: What are the differences in binding targets and their expression? ChIP and RNA-seq assays should be performed. Are there any common targets such as p21? These are important unbiased approaches. Also, what about metabolomics or RNA-seq of the bigger and fattier tumors compared to those derived from wt mice? Both analyses could unveil important clues to support mechanistically the oncogenic role of TAp73γ

6. Figure 1 legend is missing D and E. In fact, C is describing panel E.

7. Figure 2A: Is there a PTC or any changes in ORF after splicing of exon 11? The authors should describe the isoforms more. Why do they think that the α isoform is expressed more than the others?

---

## [Author Response]

Essential revisions:1. It appears unclear why the authors focus their analysis on a relatively small number of non-tumoral and tumoral prostate tissues. Did they analyze matched non-tumoral vs tumoral tissues? A table resuming the clinical features of the analyzed patients' needs to be included. Do the anyalzed samples present loss of exon 11?

We thank the reviewer’s comments. We would like to note that due to limited access to tissues, the number of samples is relatively small and the clinical features for these samples are not available. However, our data indicated that due to loss of exon 11, p73γ is highly expressed in prostate tumor tissues when compare to normal tissues. These data are still advisable for future studies with a larger sample size.

2. Interestingly, the authors reported that "E11+/- mice were bigger and fattier as compared to WT mice" with evident signs of liver steatosis. These findings might be, at least in part, mechanistically supported by TAp73γ-induced Leptin expression. A number of experiments in vitro and in vivo have been elegantly performed to show altered expression of the axis TAp73γ/Leptin in H1299 and MiaPaCa2 engineered cell lines. While these findings can be mechanistically provocative there is no any link with NSCL and prostate cancers in humans. Is Leptin expression altered in NSCL and prostate cancer databases? This is critical to translate the reported findings into a realistic cancer context. An experimental effort on this direction should be included.

We thank the reviewer’s comments. Several reports have shown that the leptin signaling pathway is elevated in several types of cancers, including lung cancer and prostate cancers (Prostate. 2001 Jan 1;46(1):62-7; Oncotarget. 2017; 8:19699-19711; Journal of Cancer Research and Therapeutics 13(2):p 204-207, 2017). However, when we tried to determine an association between p73γ and Leptin by analyzing the prostate cancer and lung cancer database from TCGA, we found that p73γ transcript was not properly annotated by GENCODE, which provide reference sequence for data analysis. Ongoing study in the lab is trying to manually match p73γ transcript in various cancer types and will be published in the future.

3. Statistical analysis is lacking throughout the manuscript. Figures 6L, 7B, 7C and supplemental Figure 5C and D do not have any statistics. The rest of the figures have asterisks but what do they mean? The asterisks are not explained and there is no footnote for p-values. Also, it is not clear sometimes how many replicates each experiment was conducted especially for migration and scratch would assays.

We thank the reviewer’s comment. Fisher’s exact test was used in the Figure 5C-D. For Figure 6L, 7B, and 7C, student *t* test was used. The asterisks indicated that p<0.05. We have modified the figure and figure legends accordingly.

4. It is not very clear why the authors chose to do semi quantitative PCR and not RT-qPCR throughput the manuscript. Given that many of the changes are quite modest (example Figure 3A), RT-qPCR should be performed for the siRNA experiments. Likewise, for the ChIP experiments in Figure 6J, ChIP-qPCR should also be performed to get a sense of the magnitude of the enrichment.

We thank the reviewer’s comment, as the regulation of Leptin by p73γ has been confirmed at both transcript and protein levels, qPCR was not performed for verification. Nevertheless, ChIP-qPCR was performed to verify the binding of endogenous p73γ to the p21 and Leptin promoter and showed that p73γ was able to bind to both promoters (please see Figure 6L in the revised manuscript).

5. What are the global transcriptional differences between TAp73 α and γ isoforms: What are the differences in binding targets and their expression? ChIP and RNA-seq assays should be performed. Are there any common targets such as p21? These are important unbiased approaches. Also, what about metabolomics or RNA-seq of the bigger and fattier tumors compared to those derived from wt mice? Both analyses could unveil important clues to support mechanistically the oncogenic role of TAp73γ

We thank the reviewer’s comments. These are important questions to elucidate the biological functions of TAp73 α and TAp73γ isoforms. However, these questions have beyond the scope of the current manuscript. Indeed, ongoing studies in the lab are trying to analyze the common and distinct functions of TAp73 α and TAp73γ by using various systems. We hope that we can report our new findings in a separate manuscript.

6. Figure 1 legend is missing D and E. In fact, C is describing panel E.

The legend for Figure 1D-E has been added. We also fixed the legend for Figure 1C.

7. Figure 2A: Is there a PTC or any changes in ORF after splicing of exon 11? The authors should describe the isoforms more. Why do they think that the α isoform is expressed more than the others?

We thank the reviewer’s comments. p73γ is produced due to splicing out exon 11. As a result, the last 76 aa in p73γ is unique. A description regarding p73α and p73γ has been added to the revised manuscript. Additionally, several studies have shown that p73α is the most commonly expressed isoforms in normal human and mouse tissues (Cell Death and Disease volume 12, Article number: 745 (2021); Cell Cycle 11:23, 4474–4483).